

# Measurement of specific surface area of fresh solid precipitation particles in heavy snowfall regions of Japan

Satoru Yamaguchi[1], Masaaki Ishizaka[1], Hiroki Motoyoshi[1], Sent Nakai[1], Vionnet Vincent[2,3], Teruo Aoki[4], Katsuya Yamashita[1], Akihiro Hashimoto[5], and Akihiro Hachikubo[6]

[1]Snow and Ice Research Center, National Research Institute for Earth Science and Disaster Resilience, Nagaoka, 940-0821, Japan

[2] Univ. Grenoble Alpes, Université de Toulouse, Météo-France, CNRS, CNRM, Centre d'Etudes de la Neige, Grenoble, France

[3] Centre for Hydrology, University of Saskatchewan, Saskatoon, Canada

[4] Arctic Environment Research Center, National Institute of Polar Research, Japan

[5] Meteorological Research Institute, Japan Meteorological Agency, Tsukuba, Japan

[6] School of Earth, Energy and Environmental Engineering, Kitami Institute of Technology, Kitami, 090-8507, Japan

*Correspondence to*: Satoru Yamaguchi (yamasan@bosai.go.jp)

**Abstract**: In countries such as Japan, particular solid precipitation particles (PP) like unrimed PP and graupel sometimes form a weak layer in snow that subsequently triggers slab avalanches. It is therefore essential for avalanche prevention authorities to design a predictive model for slab avalanches triggered by weak PP layers. The specific surface area (SSA) is a parameter that could characterize the physical properties of PP. The SSAs of solid PP were measured for four winters (from 2013/2014 - 2016/2017) in Nagaoka, one of the heaviest snowfall-experiencing cities in Japan. More than 100 SSAs of PP were measured during the study period using the gas absorption method. The measured SSA values range from 42 to 153 $m^2\ kg^{-1}$. PP under the melting condition show smaller values than that without melting. Unrimed and slightly rimed PP exhibited low SSA, whereas heavily rimed PP and graupel exhibited high SSA. The degree of riming depends on the synoptic meteorological conditions. Based on the potential of weak PP layer formation with respect to the degree of riming of PP, the results indicate that SSA is a useful parameter for describing the characteristics of PP to predict avalanches triggered by weak PP layers. The study found that the values of SSA strongly depend on wind speed ($WS$) and wet-bulb temperature ($Tw$) on the ground. SSA increases with increase in $WS$, and decreases with increase in $Tw$. Using $WS$ and $Tw$, an equation to empirically estimate the SSA of fresh PP in Nagaoka was established. The equation successfully reproduced the fluctuation of SSA. The SSA equation, with the meteorological data, is an efficient first step toward describing the development of weak PP layers in the snow cover models.





## 1. Introduction

Individual snow crystals are made of ice structures with unique, intricate geometries (Magono and Lee, 1966). The specific surface area (SSA) of snow is defined as its surface area per unit mass or volume, and includes information on the size and shape of the snow particles. Therefore, SSA is a key parameter in understanding the exchange of matter and energy between a snow-covered surface and the atmosphere (Domine et al., 2006; 2007; 2008) and in modeling the mass transfer of air or water in snow (e.g. Arakawa et al., 2009, Colonne et al., 2012). For these reasons, temporal variations in the SSA of snow cover are important to accurately simulate the physical properties of snow. Several studies (Legagneux et al., 2003; 2004, Legagneux and Domine, 2005, Taillandier et al., 2007) have proposed empirical equations for the time variation of SSA, in which an initial value of SSA was used. Therefore, the initial values of SSA, namely SSA of fresh precipitation particles (PP), are essential to simulate the time variation of SSA in natural snow cover.

PP sometimes form a weak layer in a snowpack, which may trigger slab avalanches (Akitaya and Shimizu, 1988, McClung and Schaerer, 1993). The formation of weak PP layers should be considered to be dependent on the degree of PP riming (Lachapelle, 1967). Construction on the degree of PP riming is hence a key factor in designing a predictive model for slab avalanches triggered by weak PP layers. To investigate the types of weak layers in surface avalanche, McCammon and Schweizer (2002) reported that the main cause of weak layer in the Swiss Alps and Canada is the recrystallization-type layer: depth hoar, faceted crystal, and surface hoar, while the main cause of weak layers of snowpack in Japan is the PP types (Ozeki and Akitaya, 1995, Matsumura, 2002, Ikeda, 2007). Therefore, forecast of weak-PP-layer-triggered avalanche is necessary for avalanche prediction in Japan.

Akitaya and Shimizu (1988) reported that the large-size falling broad-branched, unrimed snow crystals can form a weak layer when they fall under windless condition because its initial density is small. From the view of physical characteristics of PP layer, several studies (Shidei, 1953, Nakamura et al., 2014, Ishizaka et al., 2018) reported that snow layers consisting of unrimed snow crystals were fragile and triggered snow avalanches, even when the weak PP layers in the snowpack were shallow. In fact, the measured repose angle with the unrimed snow crystals were 35–45° (Ishizaka et al., 2018; Kamiishi et al., 2016), which are much smaller than those of the rich-rimed snow crystals (> 90°) reported by Narita and Takeuchi (2009). These results indicate the potential relationship between avalanche behavior and the weak PP-layer characteristics. However, the dependence of avalanche characteristics on PP which are necessary in predicting the potential of a weak PP layer to trigger an avalanche is still debated because previous studies could not gather any objective physical information on PP, such as shape, size and riming ratio. To solve this problem, the SSA of snow is considered an ideal parameter to form detailed information of PP.

Several studies in the past have reported SSA measurements of fresh snow (Legagneux et al., 2002, Cabanes et al., 2002, Domine et al., 2007, Schleef, 2014). Their measurement interval after the snowfall showed wide variations from several hours to a day. In general, SSA decreases with time due to metamorphism (Legagneux et al., 2003; 2004, Cabanes et al., 2003,



Legagneux and Domine, 2005, Taillandier et al., 2007). Kerbrat et al. (2008) implied that smoothing of the snow crystal's surface due to the Kelvin effect is very fast, i.e., surface roughness of the size of several micrometers disappears after a day, even at temperatures as low as -40 °C, and the smoothing accelerates with increase in the temperature. In general, the size of a riming droplet (cloud drop) is in the order of one to ten micrometers, as reported by Harimaya, (1975) and Mosimann et al.

(1994). Therefore, SSA measurement within short intervals is needed to model the degree of riming of PP, especially in warmer environment, which is common in Japan. Snow-cover models, which are used for avalanche forecasting (e.g. Brun et al., 1992, Lehning et al., 1999), generally require input data with high time resolution (e.g. 1 h) for avalanche forecasting. Therefore, if information on PP is to be introduced in snow-cover models using the SSA, then a dataset of SSA information on fresh PP with high time resolution must be developed. Moreover, this information should contribute to improve SSA of PP treatment

in the snow-cover models, which are basically simplified in the model, e.g., the Crocus snowpack model (Brun et al., 1992; Vionnet et al., 2012) treats that the maximal value of SSA of PP is 65 $m^2$ $kg^{-1}$ for low wind conditions and it decreases with increasing wind speed down to 25 $m^2$ $kg^{-1}$ (Vionnet et al., 2012, Carmagnola et al., 2014).

The SSA measurements of fresh PP were conducted over short intervals in an area in Japan that receives heavy annual snowfall, and the measurements were compared to meteorological data and detailed falling snow data for discussing the

dependency of SSA on these data. Based on these analyses, an empirical equation to estimate the SSA of fresh PP using surface meteorological data was proposed for introducing detailed information on PP to the snow metamorphism models.

## 2. Methodology

### 2.1. Observation site

The SSA observations were conducted over four winters from 2013/2014 to 2016/2017, at the Snow and Ice Research Center (SIRC), National Research Institute for Earth Science and Disaster Resilience (NIED), Nagaoka, Japan (37°25′N; 138°53′E; 97 m a.s.l.). The SIRC is located in a coastal region facing the Sea of Japan, where strong northwesterly monsoons blow from Siberia to the Japanese islands, accumulating large amounts of water vapor when passing over the warm sea and bringing heavy snowfall. The climatic condition considered at the SIRC includes a typical maximum winter snow height greater than

1.4 m, but a mean daily winter air temperature (Dec–Feb) greater than 2 °C (Yamaguchi et al., 2018). Various types of solid PP (snow flake, graupel, riming crystal, unrimed crystal, melting snow crystal, sleet, etc.) appear at the SIRC and the type of PP frequently changes within a short interval due to the change of precipitation mode (Ishizaka et al., 2013).

The Falling Snow Observatory (FSO) (Fig. 1a) at the SIRC has a cold room (-5 °C) with a 1.2 × 0.6 m roof opening (Fig. 1b) (Ishizaka et al., 2013; 2016). This setup allows the accumulation of falling snow on a flat table in the cold room under

windless condition (Fig. 1c). The PP photographs in winter are automatically captured in the cold room using a belt conveyer system (Fig. 1d). Additionally, the characteristics of falling snowfall particles, including size and fall speed, are automatically measured using a CCD camera system (Ishizaka et al., 2004). Using these characteristics, Ishizaka et al. (2013) presented a new parameter that quantitatively describes the main types of snowfall hydrometeors and reflects the contribution of all



hydrometeors to precipitation. In their method, the dominant snowfall type was represented by a pair of characteristics, size and fall speed, which were obtained from the average size and fall speed and weighted by the mass flux of all measured hydrometeors. This is termed the center of mass flux (CMF) distribution. Because size-fall speed relationship of hydrometeors is a good representation of particle types, the dominant snow type in a snowfall event may be deduced from the location of the

CMF in size-fall speed coordinates. Based on the concept of CMF, Ishizaka et al. (2016) also established approximated relationships between CMF density and initial density. The authors of the current paper used the detailed characteristics of falling snow produced by the CMF when discussing the relationship between the measured SSA of fresh PP and the characteristics of falling snow (Section 3. 4).

       The SIRC also acquires standard meteorological measurements (air temperature and relative humidity at 3.5 m above ground

level, wind speed and wind direction at 8.7 m above ground level, precipitation at 3.1 m above ground level, incoming and outgoing shortwave/longwave radiations at 6.5 m above ground level, air pressure measured at 3.0 m above ground level, and surface temperature, snow height, and snow water equivalents) at various time resolutions (1 min, 10 min, and 1 h) on field (Yamaguchi et al., 2018). These meteorological data were used with a 1-min resolution for conditioning the SSA data (Section 3.2) and use the same for discussions on the relationship between the measured SSA of fresh PP and near-surface

meteorological data (Section 3.5). This study uses these meteorological data with a 10-min resolution for estimating the SSA (Section 3.6). In addition to these measurements, the SIRC also operates a Doppler radar on the rooftop of its building and gathers information on snow clouds information during the winter season (Nakai et al., 2019). This study used the SIRC's Doppler radar data when discussing the relationship between the SSA and radar echo patterns relating to the development of snow clouds (Section 3. 3). In addition to these data, weather charts produced by the Japan Meteorological Agency

(http://www.data.jma.go.jp/fcd/yoho/hibiten/index.html) were used when discussing the synoptic meteorological condition (Section 3. 3).

## 2.2. Measurement of specific surface area of fresh PP

       The methane gas adsorption method (Domine et al., 2001, 2007; Legagneux et al., 2002) was used to measure the SSA of

fresh PP. Principles of the methane gas adsorption method have been described in detail by Legagneux et al. (2002). Recently, a portable device was developed for the methane adsorption method (Hachikubo et al., 2014; 2018). This device allows to measure the SSA of snow within a 1 h resolution (measurement repeatability (standard deviation) of 3%) (Hachikubo et al. 2012; 2013). The characteristics of this device make it very convenient for use in the current study of the SSA of fresh PP with a short interval measurement. The samples used in this study were made of fallen snow deposited within 1–2 h on a table in a

cold room (-5 °C) at the FSO. The new device requires a 30-mL sample for each measurement. The samples were directly taken from the snow deposited on the table during the measurement interval, but the PP gathered by a broom were added to the sample when the snow deposited on the table did not have enough height. Because of the short deposition time under the cold-room condition, the effect of metamorphism on the sample, which requires longer deposition time, was neglected in this



study. The measured SSAs in this study were calculated by dividing the measured surface area of the sample by its mass, which was in turn measured by an electronic balance with a 0.01-g resolution. Therefore, its unit is m$^2$ kg$^{-1}$. The microphotographs of the sample were manually captured to determine the crystal types.

## 3. Results and discussion

### 3.1 Measured SSA of fresh PP

A total of 102 SSA measurements were collected from the samples acquired over four winters (from 2013/2014 to 2016/2017). The averaged heat of adsorption, which is an indicator of the judgment of measurement quality, for the 102 measurements was 2472 ±199 J mol$^{-1}$, which was consistent with the value of 2540 ±200 J mol$^{-1}$ recommended by Domine et al. (2007). Therefore, the measurement results stand to reason. Figure 2a shows the comparison results between our results and SSA of natural freshly fallen snow in the previous studies (Domine et al., 2007, Schleef, 2014) (Hereafter, data of Domine et al. (2007) is referred to as Dom2007 and that of Schleef (2014) as Sch2014). The figures also show the optical radius ($R_{opt}$) estimated from the SSA using the following equation:

$$R_{opt} = \frac{3}{SSA \times \rho_i} \qquad (1),$$

where, $\rho_i$ is the density of ice (917 kg m$^{-3}$).

The sample numbers of the SSA measurements in this study are larger than those in the previous studies, and they present a wider range of SSA values. This could be explained by the more diverse variety of PP types in this study. The average value of data in this study (96 m$^2$ kg$^{-1}$) is larger than those in previous studies (Dom2007: 73 m$^2$ kg$^{-1}$, Sch2014: 79 m$^2$ kg$^{-1}$). Fassnacht et al. (1999) simulated the amount of change in the SSA of a dendritic snow crystal caused by the presence of riming drops on the surface. They concluded that the SSA could be doubled if 20% of the surface of the snow crystal were covered by needle- or plate-shaped rime. Therefore, one of the reasons the averaged value in this study is larger is the measured data of fresh PP includes many cases of graupel and rich-rimed snow crystals. Another reason could be the condition of the samples. The samples in this study were measured within a short time (1–2 hours) after deposition, so the effect of metamorphism should be small. On the other hand, the samples in the previous studies were procured at a longer span after deposition. Thus, the effect of metamorphism on the samples in those studies should be larger than on the samples in this study. For these reasons, the results of this study show a more realistic value of the SSA of fresh PP.

Large variations in the measured SSA values were observed in every fourth season (2013/2014: 64 –153 m$^2$ kg$^{-1}$, 2014/2015: 43–142 m$^2$ kg$^{-1}$, 2015/2016: 42–148 m$^2$ kg$^{-1}$, 2016/2017: 51–110 m$^2$ kg$^{-1}$) (Fig. 2b). These results indicate that the SSA of fresh PP in the winter season in Nagaoka usually varies by more than three-fold. To evaluate the effect of fluctuation in SSA of fresh PP on surface albedo, surface albedos in the visible spectral region and the near-infrared spectral region were simulated using the measured maximum (77 μm) and minimum optical radii (22 μm) of fresh PP (Fig. 2), via the "physically based snow albedo model" (PBSAM) developed by Aoki et al. (2011). These simulations were conducted for the case of impurities-free snow under a clear sky condition at a solar zenith angle of 60˚. Previous studies (Wiscombe and Warren, 1980,





Aoki et al., 2011) show the albedo at the NIR wavelengths is affected by the change of SSA more significant than that for the visible. In fact, the visible albedo value simulated using the measured maximum and minimum optical radii show almost the same values (0.99), while the simulated near-infrared (NIR) albedo values vary from 0.75 to 0.80, resulting from the SSA range of fresh PP. These results indicate that the information on SSA variation of fresh PP should be important for the
simulation of the NIR albedo.

**3.2 Influence of melting effect in SSA of fleshly PP**

According to Yamaguchi et al. (2013), winter precipitation in Nagaoka frequently occurs at an air temperature of approximately 0 °C. In fact, the data in this study was sometimes measured at an air temperature of approximately 0 °C. Thus,
some of the samples were affected by melting during the fall. To investigate the influence of melting on the measured SSA data, the measured data was classified into two categories—data without melting effect (*no melt events*) and data with melting effect (*melt events*). To classify these data, the wet-bulb temperature (*Tw*) during the falling snow period was used, which is a good indicator of the melting event (Matsuo et al., 1991); *no melt events* followed $Tw < 0$ °C condition, while *melt events* followed $Tw \geqq 0$ °C condition. In this study, the period with falling snow during the sample period was first determined
using the CMF data with a 1-min time resolution, because falling snow did not always occur continuously during the sample interval. Then, using the meteorological data also with 1-min time resolution, all the relevant meteorological elements were averaged over the period in which snowfall was observed, instead of averaging over all periods of the sample interval for calculation (Hereinafter, the averaged meteorological data will indicate the data averaged only over the period of snowfall). In this study, *Tw* was calculated based on the "forward" analytical psychrometric equations (Bohren and Albrecht, 1998) that
employ an iterative approach using averaged air temperature, relative humidity, and air pressure.

Figure 3 shows the selected results using *Tw*. Thirty events were classified as *melt events* while 72 events were classified as *no melt events*. The SSA averaged over the *melt events* (77±23 m$^2$ kg$^{-1}$) is smaller than that of the *no melt events* (103±28 m$^2$ kg$^{-1}$). Here the values after ± represent one standard deviation. The difference in the SSA values between *melt events* and *no melt events* should be mainly caused by the melting effect. Domine et al. (2007) reported an SSA value of 50±11 m$^2$ kg$^{-1}$
for fresh PP with melting effect, i.e., PP falling under T > 0 °C. Compared to the results of this study, the values in their study are smaller. Moreover, the SSA data under *melt events* in this study include higher values (>100 m$^2$ kg$^{-1}$). Figure 4 shows the microphotographs of the two samples taken under *melt events* with different *Tw*. The first sample, which showed evidence of melting under high *Tw* (0.6 °C), had a small SSA (51 m$^2$ kg$^{-1}$) (Fig. 4a) while the second sample, which was taken under *melt event* with low *Tw*, slightly higher than 0 °C, did not reveal any substantial evidence of melting and had a large SSA (113 m$^2$
30  kg$^{-1}$) (Fig. 4b). These results indicate that the degree of melting depends on *Tw*. Moreover, these also imply that the classification using *Tw* is not sufficient to distinguish between melt and not melt. Basically, data under *no melt events* show high SSA values but still include lower SSA values (< 77 m$^2$ kg$^{-1}$). These results indicate that fresh PP can have small an SSA without melting, and other factors for controlling SSA of fresh PP ought to be considered.



### 3.3 Relationship between SSA of fresh PP and synoptic meteorological condition

The measured SSA of fresh PP sometimes reveals small values without melting (Fig. 3). To investigate the cause of these small SSA values, the study will focus on the synoptic meteorological conditions during snowfall. Snowfall patterns in Japan can be roughly grouped into two categories (Nakamura et al., 1987): monsoon-type snowfall, in which strong northwesterly monsoons blow from Siberia to the Japanese islands, and cyclone-type snowfall, in which cyclones blow from the south to the north along the archipelago. The first type was named "M-type" (Fig. 5a) and the second "C-type" (Fig. 5b). In Nagaoka, snowfall events under the M-type are dominant, while snowfall events under C-type occurs rarely during winter season (Nakamura et al., 1987). Based on classification using the weather charts produced by the Japan Meteorological Agency (http://www.data.jma.go.jp/fcd/yoho/hibiten/index.html), sixty events resulted from M-type and 12 events from C-type in the no melt data with 72 events. Figure 6 is the result of comparison of the SSA values between M-type and C-type. There is a clear difference in the SSA values between M-type and C-type: SSAs during M-type (average: 112 $m^2$ $kg^{-1}$) are much larger than those during C-type (average: 60 $m^2$ $kg^{-1}$). These results indicate that the SSAs of fresh PP in Nagaoka strongly depend on the synoptic scale precipitation condition. In general, the PP during the M-type pattern generally allows aggregation to be predominant, as well as riming (Fig. 5c). Previous study (Cabanes et al., 2002) reported dendritic snowfall with rich riming to show the high SSA in the Canadian Arctic.

Colle et al. (2014) reported that fallen snow during the cyclone at Stony Brook in New York on the northeast coast of the United States showed consistent spatial patterns of habit and riming intensity relative to the cyclone structure. Little to no riming snowfall crystals occurred within the outer comma head to the north and northeast of the cyclone's eyes and the western quadrant of the comma head, while moderately rimed snow crystal were observed in the middle of the comma head. In Japan, several studies with meteorological condition analyses (Nakamura et al., 2013, Akitaya and Nakamura, 2013) reported that unrimed snow crystals fell at the warm front of the cyclone and subsequently caused avalanches. These previous studies corroborate each other, namely, unrimed snow crystal occurred with the cyclone event. In fact, all observed crystals under C-type showed the unrimed snow crystal types shown as in Fig. 5d. From these results, it was concluded that the characteristics of PP under C-type, namely unrimed snow crystal types, reveal PP with small SSA. Moreover, these results show the potential of SSA to describe riming condition of PP.

Several disastrous avalanches have recently occurred in Japan: In February 2014, hundreds of avalanches occurred simultaneously in the Kanto region, which typically receives limited snowfall during the winter. Because of the risk of avalanche, many villages, with over 9,000 residents were cut-off from access to roads and service (Nakamura et al., 2014, Kamiishi and Nakamura, 2016). In March 2017, eight people, including seven high school students, lost their lives in an avalanche during an extracurricular outing (Nakamura et al., 2017, Araki, 2018). All of these avalanches were caused by the weak PP layer resulting from unrimed snow crystals under C-type conditions. To predict this type of avalanche, introduction of information of PP type to numerical snow models is essential for reproducing weak PP layer, this study considered the



riming condition of PP determined based on SSA value to be a good indicator for the potential of weak PP layer development in snowpack for numerical snow cover models.

To investigate the measurements under M-type, SSA values of fresh PP still show a large variation in the range of 64 m² kg⁻¹–154 m² kg⁻¹. Nakai et al. (2005) reported that there were several snowfall conditions corresponding to different snow cloud behaviors under the M-type pattern. They investigated snow cloud behavior (snowfall modes) at the Niigata Prefecture, including Nagaoka, using Doppler radar echo under M-type, and then classified the snowfall modes according to six modes (Table 1). Based on their method, in this study, data under M-type were classified into several snowfall modes. Fifty-three out of 60 sets of data under M-types were classified into four modes (T mode: 20 events, L model: 24 events, D mode: 4 events, S mode: 5 events), while seven sets of data could not be undetermined because of lack of radar data or difficulty in classification resulting from the complexity snow mode. In this study, SSA under the V mode and the M mode were not measured. Figure 7 shows the results of the SSA for each snowfall mode classification. For more information, the sample microphotographs of PP taken under each mode are also shown in Fig. 8. The values of SSA under the T and L modes show larger values (averaged SSA values of the T and L modes are 120 and 119 m² kg⁻¹, respectively) than the other two modes. The previous study (Harimaya and Nakai, 1999) reported that falling snow crystals contain rich-rimed types under the T and L modes. As a matter of fact, the main types under the T and L modes in this study were the graupel and rich-rimed snow crystals (Fig. 8a and 8b). The measured values of SSA under the D mode (averaged SSA value of the D mode is 93 m² kg⁻¹) show smaller values than the T and L modes, but their values showed large fluctuations from 60–120 m² kg⁻¹. This result implies that various PP types, from slightly rimed snow crystal to rich-rimed snow crystal, should fall under the D mode. The microphotographs in Fig 8c show that different crystal types can fall under the D mode. Because of the small sample size of the D mode (only four cases), the critical condition between the PP fall with a large SSA and that with a small SSA could not be established in this study. Although slightly larger than the values under C-type, the measured values of SSA under the S mode (averaged SSA value of the S mode is 75 m² kg⁻¹) show smaller values than the other modes. Microphotographs of all samples under the S mode reveal unrimed and slightly rimed snow crystals as the classification of the degree of riming for single crystal (Mosimann et al., 1993) (Fig. 8d). The sample size of the S mode (only five cases) does not help to reveal if all the cases under this mode have small SSA values, but the results do indicate that the PP with small SSA (unrimed or slightly rimed snow crystals) appeared even under M-type. Therefore, the risk of avalanche caused by the weak PP layer, resulting from unrimed snow crystals fallen should not be neglected even under M-type.

### 3.4 Relationship between SSA of fresh PP and detailed characteristics of PP produced by CMF analyses

In this section, the relationships between the SSA of fresh PP and their characteristics are discussed in detail. The PP type sometimes varies even for observation periods (1–2 h) (Ishizaka et al., 2016): therefore, it was necessary to select simpler cases with the same PP type during the sample deposition period, to clarify the relationship between the SSA of fresh PP and its characteristics. To examine the quality of PP type variations, Ishizaka et al. (2016)'s CMF distribution during the sample



deposition period was considered. In the analyses, CMF was averaged over the sample deposition period (averaged CMF) and over each 1-min interval (1-min CMF). The pattern of a 1-min CMF can remain unchanged even as the particle size differed during the sample deposition period (Fig. 9a). In such a case, these events were regarded as "*uniform falling events* (UFE)," which have a single PP type during the deposition period. On the other hand, the pattern of a 1-min CMF can also fluctuate in

size-fall speed coordinates, while the averaged CMF is located in an intermediate area during the sample deposition period (Fig. 9b). In this case, these events were regarded as "*variant falling events*," which have mixed PP types during the deposition period. CMF distributions of all cases were graphed and inspected visually based on these analyses, forty-nine UFE were selected. Figure 9 shows the results of comparison between data of *no melt events* (72) and data of UFE (49). Although the number of UFE became two-thirds of the number of *no melt events*, its dispersion still follows a trend similar to that of *no melt*

*events*. For this reason, hereinafter only data of UFE will be used for detailed analyses.

As shown in Fig. 6, the SSA strongly depends on its synoptic scale condition and the relationship between SSA of fresh PP and detailed characteristics of PP may also depend on its synoptic scale condition. Therefore, firstly the UFE data were classified into the two synoptic scale conditions (M-type and C-type). In addition, M-type data were classified into three groups based on the PP types (aggregate group (A), graupel group (G), and small particle group (S)) using CMF analyses reported by

Ishizaka et al. (2016) (Fig. 11). Although Ishizaka in their paper divided a small particle group (S) into two subgroups (S1 and S2), the authors in this study treated S1 and S2 as one group (S) (Fig. 11).

Figure 12 shows the relationship between the SSA of fresh PP and detailed characteristics of the PP produced by the CMF analyses for each type. In this study, three physical characteristics obtained from CMF analyses were adopted— averaged fall speed ($V$) of PP and averaged apparent size ($D$) of PP. These two elements were determined by the CCD image for each particle

and then averaged over all the particles during the sample deposition period (Ishizaka et al., 2013). Third is the initial deposited density ($Ro$), which was calculated based on five-min averaged CMF data using the method of Ishizaka et al. (2016). Though $V$ and $D$ were directly measured, the initial deposited densities were estimated value using $V$ and $D$.

In the case of C-type, significant correlations (p value (p) is smaller than 0.05) between the measured SSA and all three physical characteristics could be obtained. Parameters $V$ and $Ro$ decreased with an increase in SSA (Fig. 12j and Fig. 12l),

while $D$ increases with an increase in SSA (Fig. 12k). The relationship between the measured SSA and $Ro$ under C-type shows a similar trend as in Domine et al. (2007).

In the M-type, including A, G, and S groups, only one significant correlation ($p \leqq 0.05$) between SSA and $D$ could be obtained in the G group. The p values for other combinations were not enough to assure significant correlations (p > 0.05). The trend between $D$ and SSA in the G group, which is a positive correlation, may result from the process of graupel growth.

It is likely that larger graupel can collect a large amount of small-rimed drops and also protect larger number of small-rimed drops in its body from sublimation during its fall. Therefore, a larger graupel stored larger number of small-rimed drops in its body. This is only an assumption, and further investigation with more measurements is needed to prove it. Furthermore, a



simulation using a detailed cloud physics model, such as Hashimoto et al. (2018a; 2018b), will equip this study with some useful information to understand the reason $D$ and SSA in the $G$ group are related.

As shown in Fig. 6, all measured SSA values of an unrimed crystal in C-type are less than 90 $\mathrm{m^2\ kg^{-1}}$ (the maximum value being 88 $\mathrm{m^2\ kg^{-1}}$). In addition, measured SSA values under the S mode in M-type, which include unrimed and lightly rimed snow crystals, are in the range of 70 to 78 $\mathrm{m^2\ kg^{-1}}$ (in the A Group). Therefore, this study simply assumes that the crystal having an SSA less than 90 $\mathrm{m^2\ kg^{-1}}$ is an unrimed crystal type or lightly rimed snow crystal, while the crystal having an SSA of over 90 $\mathrm{m^2\ kg^{-1}}$ is a rimed crystal; densely rimed, graupellike, and graupel, as the classification of the degree of riming for a single crystal (Mosimann et al., 1993). In fact, a great majority of SSAs of graupels (G group), which have a large number of rimed drops in them, are larger than 90 $\mathrm{m^2\ kg^{-1}}$ (Fig. 12d, e, f). Moreover, SSAs in the S group, which should be made of small ice crystals that may have aggregated droplets, are also mostly larger than 90 $\mathrm{m^2\ kg^{-1}}$ (Fig. 12g, h, i). To investigate A group from this perspective, namely, PP type in the A group changed from unrimed or lightly rimed snow crystals to densely rimed, graupelike, and graupel at the border of 90 $\mathrm{m^2\ kg^{-1}}$, the relationships between SSA and $D$ and $Ro$ change trends at the border of 90 $\mathrm{m^2\ kg^{-1}}$ (dotted lines in the figures); $D$ increases (decreases) with SSA when SSA < 90 $\mathrm{m^2}$ $\mathrm{kg^{-1}}$ (SSA $\geqq$ 90 $\mathrm{m^2}$ $\mathrm{kg^{-1}}$) and $Ro$ decreases (increases) when SSA increase under SSA < 90 $\mathrm{m^2\ kg^{-1}}$ (SSA $\geqq$ 90 $\mathrm{m^2\ kg^{-1}}$). Moreover, the relationships between SSA and $V$ and $Ro$ of A group under SSA < 90 $\mathrm{m^2\ kg^{-1}}$ are similar to those of C-type. These results indicate that the threshold value of 90 $\mathrm{m^2\ kg^{-1}}$ between the unrimed and rimed crystals, which were determined roughly, is adequate from the perspective of characteristics $V$ and $Ro$. Moreover, there is possibility that the SSA of PP closely depends on their physical characteristics (mainly degrees of riming and size for unrimed PP), and thus the physical characteristics of PP are treated as a function of SSA.

**3.5 Relationship between SSA of fresh PP and meteorological data on ground**

The SSA of PP should be governed by synoptic conditions and meteorological conditions in the clouds, but meteorological conditions in the clouds cannot be fully characterized at the present. The parameterization of SSA of fresh PP should be useful to introduce the fresh PP characteristics for the snow-cover models, which are essential to predict the avalanche resulting from the weak PP layers. A previous study (Domine et al., 2007) concluded that the relationship between measured SSA of fresh PP and meteorological data on ground was not legible in their data. One of the reasons of their conclusion should originate from their PP samples taken from different sites characterized by different snow climate and synoptic condition; therefore, their SSA values should include the influence resulting from different environments. Second reason should originate from various sample measurement intervals, which were longer than those in this study; therefore, their fresh PP samples should have change due to metamorphism before the measurements, and the metamorphism degrees varied depending on their measurement interval. Moreover, meteorological conditions and falling snow types should have varied during the sample deposition because of their longer interval. From these reasons, their conditions may complicate the understanding of the relationship between SSA of fresh PP and the meteorological data on ground. On the other hand, in this study, the SSA of



fresh PP were measured at the same place with a short time interval (1–2 h), to avoid the effect of metamorphism before the measurement and the effect of fluctuation in the meteorological conditions. Therefore, in this section, the possibility of parameterizing the SSA of fresh PP using the meteorological data on ground was explored. As mentioned in Section 3.4, the PP type sometimes varied even for short observation period (1–2 h) (Ishizaka et al., 2016); therefore, only the UFE data were

used for the analyses, to focus on the relationship between the SSA of uniform PP type and meteorological condition. The following elements, averaged only over the period of snowfall (see Section 3.2), were used—air temperature ($Ta$), relative humidity ($RH$), wind speed ($WS$), air pressure ($Press$), and wet-bulb temperature ($Tw$) calculated using $Ta$, $RH$, and $Press$. The reasons for selecting these meteorological elements are that $Ta$, $RH$, $WS$, and $Tw$ are often used for the parameterization of new snow density, as summarized by Helfricht et al. (2018). Although $WS$ is also strongly affected by the local topography

and roughness, $WS$ and $Press$ should share a relation with the synoptic scale condition.

Figure 13 shows the correlations between the measured SSA and each meteorological variable ($Ta$, $RH$, $WS$, $Press$, and $Tw$). Although all variables show significant correlation with measured SSA (P value < 0.05), the relationships vary with each variable. A strong, positive relationship between the measured SSA and $WS$ was obtained with a high correlation coefficient, $R$ ($R = 0.74$); SSA increases with increasing $WS$ (Fig. 13d). The result of this study corresponds to the results of the previous

study (Harimaya and Nakai, 1999), in which the mass proportion of rime reportedly increased with an increase in the wind speed under the T and L modes. $Tw$ shows a strong negative correlation with SSA ($R = -0.60$); SSA decreases with an increase in $Tw$ (Fig. 13e). $T_a$ and $RH$ show relatively low negative correlations ($Ta$: $R = -0.44$, $RH$: $R = -0.48$) (Figs. 13a and 13b). On the other hand, the relationship between SSA and $Press$ shows a small negative correlation ($R = -0.30$) (Fig. 13c). Based on these analyses, $WS$ and $Tw$ were selected for the parameterization of SSA of fresh PP, because of their high $R$ values. Moreover,

$Tw$ indirectly involves the influences of $Ta$, $RH$ and $Press$ fluctuations because $Tw$ is calculated using those parameters. To deduce an estimation equation for SSA using $WS$ and $Tw$, SSA was assumed to be linearly dependent on $WS$ and $Tw$. Equation (2) is this parameterization:

$$SSA = 17.6WS - 9.4Tw + 58.5 \quad (R = 0.81) \qquad (2)$$

Here, the valid range of Eq. (2) is from 0–4 m s$^{-1}$ for $WS$ and from -4–0 °C for $Tw$.

Figure 14 shows the comparison between the measured SSA and the calculated SSA using Eq. (2) with $WS$ and $T_w$. The residual standard error is 16.8 m$^2$ kg$^{-1}$. These results indicate that the SSA of fresh PP can be calculated using meteorological data of Nagaoka. Although the equation includes the limitation of its parameterization, which is strongly site-specific, especially due to the introduction of wind speed in the parameter, this idea is the first step to introducing falling snow crystals into the snow cover models using SSA.

### 3.6 Calculation of time evolution of SSA of fresh PP

Here, the time-evolution estimation of the SSA of fresh PP is presented using Eq. (2), for the period of Jan 27, 2015 to Feb 1, 2015, in which the synoptic meteorological conditions periodically changed (Fig. 15). Figure 16 shows the meteorological



conditions (*WS*: wind speed, *Tw*: wet-bulb temperature, HS: total depth of snow cover, and P: precipitation) during this period. Calculated values of the SSA of Fresh PP using Eq. (2) with 10-min meteorological data (*WS* and *Tw*) are also shown in Fig. 16. In this figure, SSA was calculated only for the condition $Tw < 0\ °C$ and $P > 0$ mm, based on the 10-min meteorological data, and were then averaged over each hour to obtain hourly data. In the figure, the ranges of fluctuations of the calculated SSA during each hour and values of the measured SSA are also shown in the figure.

On 27 January, the synoptic precipitation condition was C-type (Fig. 15a). Although the precipitation had occurred, temperature condition was above 0 °C and most of the precipitations should have been in the liquid. In fact, the HS did not increase on Jan 27. The synoptic meteorological condition changed from C-type to M-type during the period of the night of 27 January to the morning of 28 January (Fig. 15b). The temperature dropped during the period, precipitation changed from liquid to solid state and the HS increased during this period. The calculated SSA gradually increased during this period (Fig. 16). The M-type condition continued from the morning of 28 January to 29 January (Fig. 15c). During this period, the temperature remained low and precipitation occurred in the solid state. HS increased continuously and the calculated SSA remained large during this period (Fig. 16). The temperature gradually increased as the cyclone lashed from the south to the north along the Japanese archipelago on 30 January (Fig. 15d). Although no precipitation occurred during the morning of 30 January, solid precipitation occurred in the evening of the same day with small values of the calculated SSA. On 31 January, the synoptic meteorological condition gradually changed from C-type to M-type (Fig. 15e). Because the temperature was high, only liquid precipitation occurred, and the HS decreased during this period. The first of February, the condition was M-type (Fig. 15f) and the temperature dropped to the low condition. Therefore, solid precipitation occurred with large values of the calculated SSA. When the calculated SSA and the measured SSA are compared, it can be seen that the former can reproduce the fluctuation in the measured SSA (SSA increased after the morning of 28 January and SSA decreased during 30 January) although absolute values shows some different between each other. As discussed in Section 3.4, if the PP with SSA < 90 m$^2$ kg$^{-1}$ were considered to be unrimed snow crystals, which should transform into a weak PP layer, the calculation results imply two possibilities (28 January and 30 January) of weak PP layer development during the study period. In fact, the authors of this study observed the unrimed PPs in Nagaoka on Jan 30, 2015 (Fig. 17). For these reasons, this study has considered that calculating the SSA using Eq. (2) with meteorological data is an efficient first step toward describing the development of the weak PP layer in the snow cover model.

## 4. Conclusions

A total of 102 SSAs of freshly PP were measured shortly after their deposition (1–2 h) at the SIRC in NIED during four winters (2013/2014, 2014/2015, 2015/2016, 2016/2017) in Nagaoka, one of the regions in Japan that receives the heaviest snowfalls and located in a coastal region facing the Sea of Japan in Honshu Island. The measured SSA values ranged widely from 40–150 m$^2$ kg$^{-1}$. To investigate the cause of variation in the SSA of fresh PP, the influence of melting during its fall was analyzed. The SSAs of fresh PP under the melting condition show smaller values than that without melting. Besides this, the



relationship between the measured SSA and synoptic meteorological conditions–monsoon-type (M-type) and cyclone-type (C-type) –was also analyzed. The measured SSAs under C-type are smaller than those of M-type, because the snow crystals under C-type are unrimed snow crystals, while those under M-type are mostly rich-rimed snow crystals and graupels. Furthermore, a detailed investigation of SSA under M-type was conducted with various snowfall modes determined using the

radar data. The results indicate the possibility that unrimed and slightly rimed snow crystals occurred at a specific snowfall mode (S mode) even under M-type. This result implies that the weak PP layer with unrimed/slightly rimed snow crystals can develop not only under C-type but also under M-type snowfalls. The analysis of comparisons between the SSA values of PP and their properties (averaged fall speed, averaged apparent size, and initial deposited density) confirms that the SSA of PP is strongly influenced by their physical properties, in particular the degree of riming.

Based on the analyses using the measured meteorological data on ground, the values of SSA were found to be strongly dependent on the wind speed ($WS$) and the wet-bulb temperature ($Tw$); SSA increases with an increase in $WS$ while SSA decreases with an increase in $Tw$. Using $WS$ and $Tw$, the equation to estimate the SSA of fresh PP was derived This equation helped simulate the fluctuation of SSA of fresh PP during the period in which the C-type and M type snowfalls appeared periodically. Thus, although it has its limits stemming from site-dependent parameterization, especially due to the introduction

of wind speed in the parameter, the equation to simulate the SSA with meteorological data is an efficient first step toward describing the weak PP layer in the new snow-cover model.

    This study focuses on estimating the SSA of fresh PPs that would help in designing a predictive model for slab avalanche caused by weak PP layer. However, to implement the development process of weak PP layer in the snow-cover model with the SSA, the dependency of physical characteristics of the PP type on its SSA values needs to be investigated in future studies.

Moreover, this study discussed the SSA of flesh PP only under one climatic condition. Additional similar measurements of SSA of fresh snow in other climatic conditions are required for understanding further the SSA of fresh snow. For further parameterizing the SSA of flesh PP, it is necessary to use the meteorological conditions of the cloud on which the SSA of fresh PP strongly depends. To promote this parameterization, further studies could compare the results of this study with detailed information on precipitation particles produced from the numerical meteorological model, such as the Japan Meteorological

Agency's non-hydrostatic model (JMA-NHM, Saito et al., 2006), with the option of double-moment bulk cloud microphysics scheme (Hashimoto et al. 2018a, 2018b) or the WRF model (Skamarock et al., 2008), which includes the P3 scheme (Predicted Particles Properties; Morrison and Milbrandt, 2015).

ACKNOWLEDGEMENT

We would like to acknowledge the members of SIRC for the useful discussion. This study was part of the project "Research on combining risk monitoring and forecasting technologies for mitigation of diversifying snow disasters". This study was also supported by JSPS KAKENHI, Grant Numbers JP23221004 (SIGMA PJ), JP15H01733 (SACURA PJ) and JP16K01340.





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



Table 1 Classification of snowfall mode (modification of Table 1 in Nakai et al (2005))

| | |
|---|---|
| L mode (Longitudinal line) | Bands running or cells aligned nearly parallel to the prevailing wind |
| T mode (Transversal line) | Bands running or cells aligned with a large angle relative to the prevailing wind |
| S mode (Spreading precipitation) | Widely spreading, relatively uniform precipitation |
| V mode (Meso-$\beta$ scale vortex) | Vortices and associated curved bands with a significant change of the wind direction |
| M mode (Mountain-slope precipitation) | Area of stationary precipitation around the windward slope of the mountains |
| D-mode (Local-frontal band) | A wide band is considered to form along a line of discontinuity |



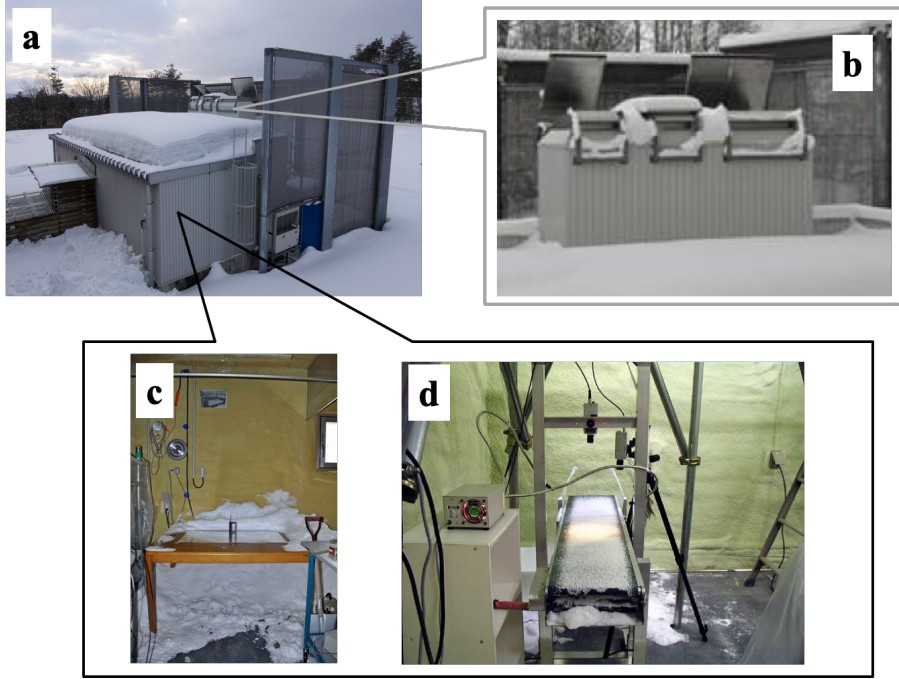

Fig.1. Falling Snow Observatory (FSO) at the SIRC
a. Overview of FSO, which is surrounded with defensive fence against wind effect
b. Roof opening system on the cold room
c. Table in the cold room, on which snow sample is naturally deposited through the roof opening
d. The system to automatically take falling snow crystal photos on the belt conveyer



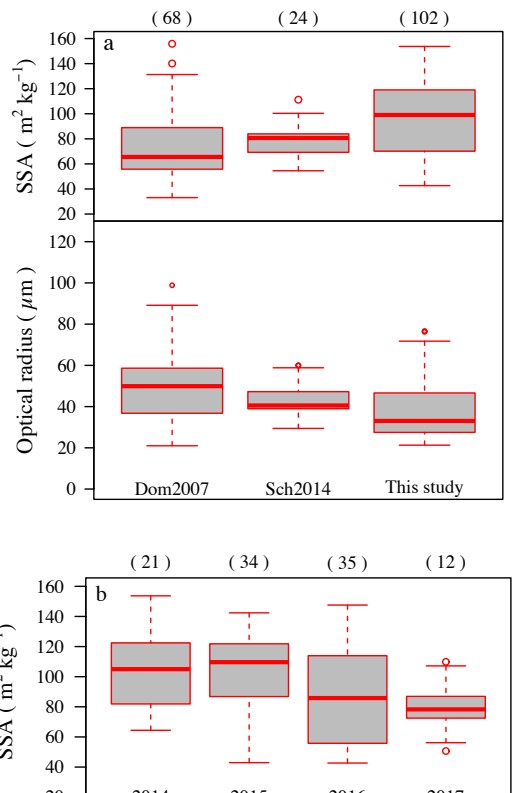

Fig. 2. Measurement results of SSA
a: Comparison between the measurement results at Nagaoka and those of fresh snow reported in previous studies.
Dom2007: data in Domine et al. (2007)
Sch2014: data of natural snow in Schleef (2014)
This study: data at Nagaoka
b: Measurement results of SSA for each year
2014: data measured in 2013/2014 winter
2015: data measured in 2014/2015 winter
2016: data measured in 2015/2016 winter
2017: data measured in 2016/2017 winter



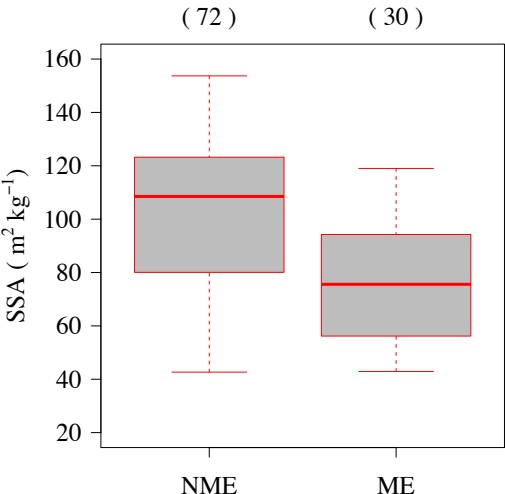

Fig. 3. SSA data classification based on the consideration of melting using wet-bulb temperature ($Tw$)
NME (*No melt events*): data with $Tw < 0$ °C
ME (*Melt events*): data with $Tw \geqq 0$ °C
Each box plot shows median, 25 and 75% percentiles, 1.5 × interquartile ranges, and outliers

Values in parenthesis are sample numbers

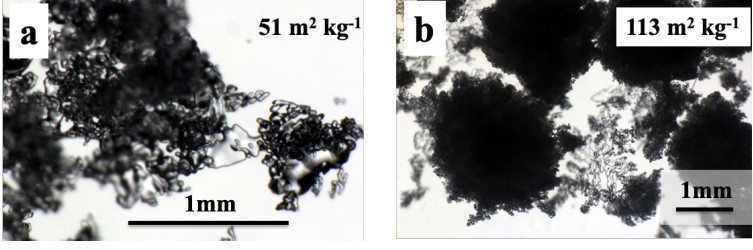

Fig. 4. Microphotographs of samples taken under melt events
a: Sample with $Tw = 0.6$ °C
b. Sample with $Tw = 0.03$ °C





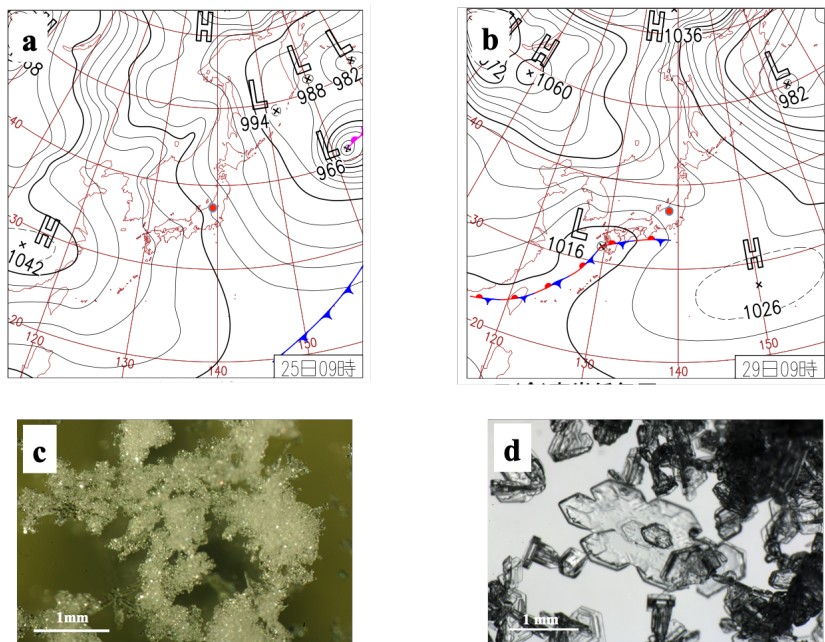

Fig. 5. Synoptic atmospheric pressure condition of falling snow events in Nagaoka
Red circles indicate the location of Nagaoka
a: Synoptic weather chart at 9:00 A.M. on 25 Jan. 2016, as an example of M-type
b: Synoptic Weather chart at 9:00 A.M. on 29 Jan. 2016, as an example of C-type
c: Microphotograph of fresh PP deposited on a table in a cold room under the condition of a
d: Microphotograph of fresh PP deposited on a table in a cold room under the condition of b
Weather charts were produced by the Japan Meteorological Agency
(http://www.data.jma.go.jp/fcd/yoho/hibiten/index.html).





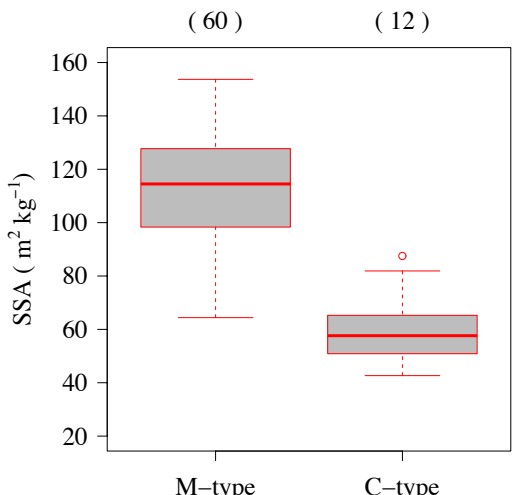

Fig. 6. Dependency of SSA of flesh PP on synoptic scale condition
M-type: Monsoon type
C-type: Cyclone type
Each box plot shows median, 25 and 75% percentiles, 1.5 × interquartile ranges, and outliers

Values in parenthesis are sample numbers




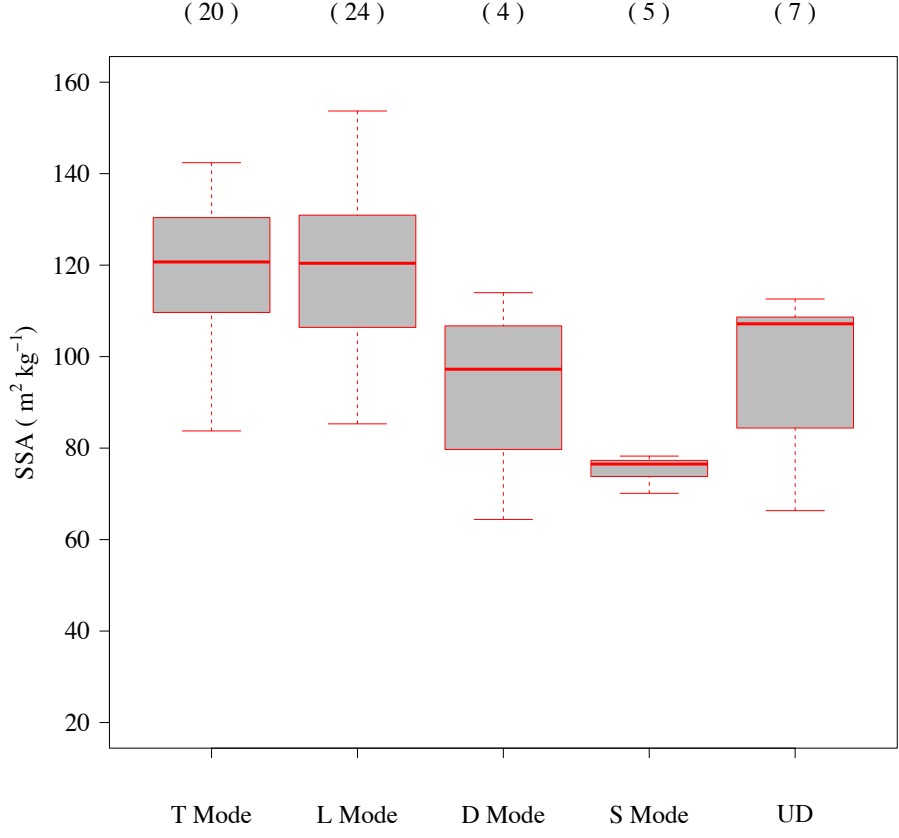

Fig. 7. Dependency of SSA of fresh PP on snowfall modes during M-types
Each mode is shown in Table 1. UD indicates data whose snowfall mode could not be determined.
Each box plot shows median, 25 and 75% percentiles, 1.5 × interquartile ranges, and outliers
Values in parenthesis are sample numbers




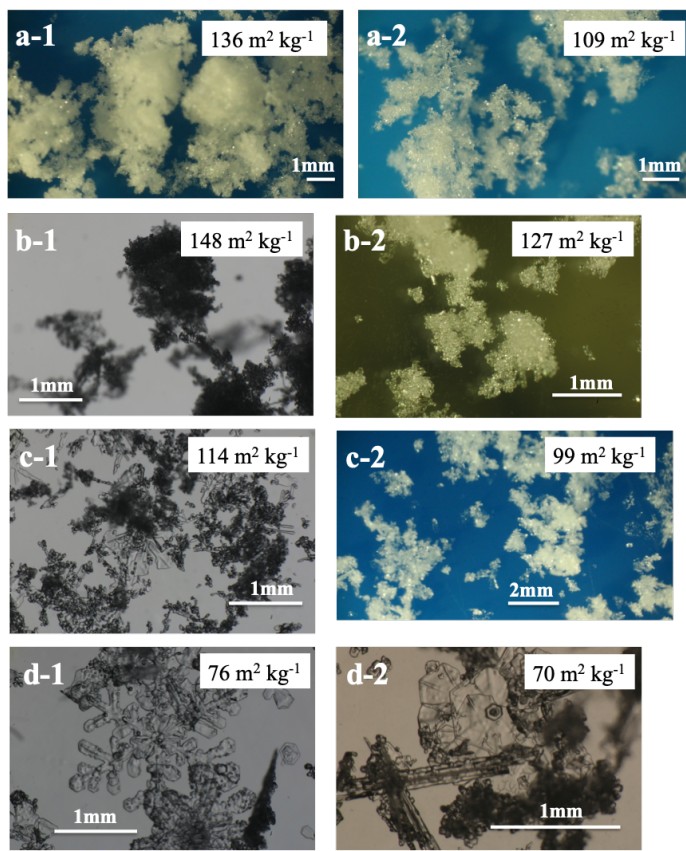

Fig. 8. Microphotographs of samples for each snowfall mode
a-1 and a-2 were taken under T mode
b-1 and b-2 were taken under L mode
c-1 and c-2 were taken under D mode
d-1 and d-2 were taken under S mode
Values in the figures are the measured SSAs.





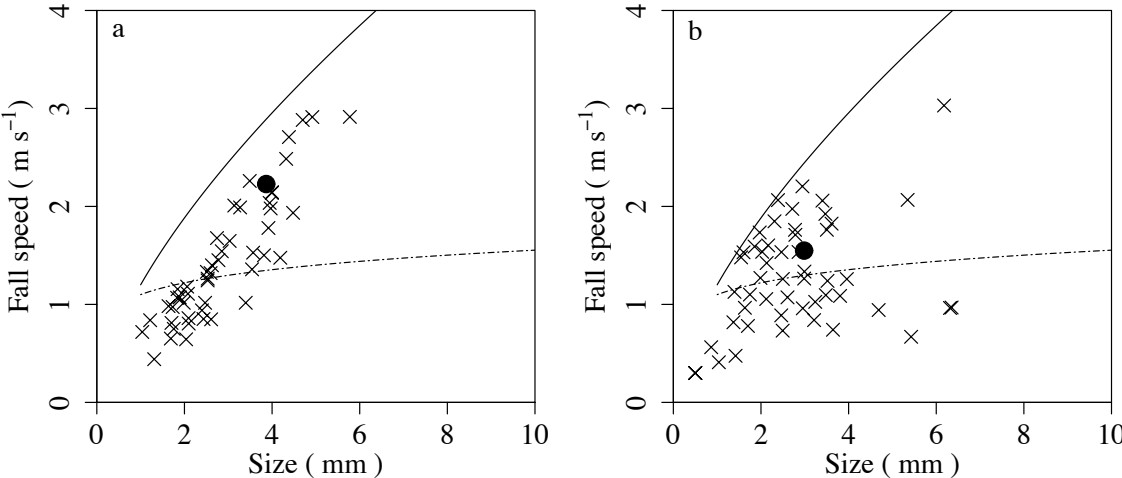

Fig. 9. The distribution of CMF with 1 min. and the integrated CMFs for events
a: Example case of *uniform falling event* (9:40–10:40 on Jan 25, 2016)
b: Example case of *variant falling event* (9:10–10:00 on Feb 10, 2015)
x-mark: CMF of each 1 min. Black circle: Averaged CMF
Solid line: conical graupel (Locatelli and Hobbs, 1974)
Broken line: Densely rimed aggregate (Locatelli and Hobbs, 1974)





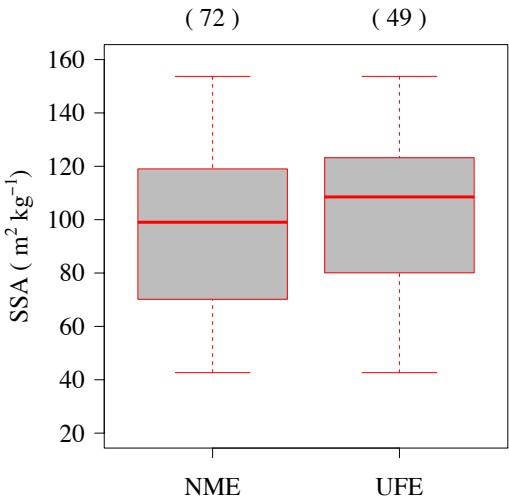

Fig. 10. Comparison between data of *no melt event* and data of *uniform falling event*
NME indicates the data of *no melt event* and UFE indicates the data of *uniform falling event*.
Each box plot shows median, 25 and 75% percentiles, 1.5 × interquartile ranges, and outliers
Values in parenthesis are sample numbers



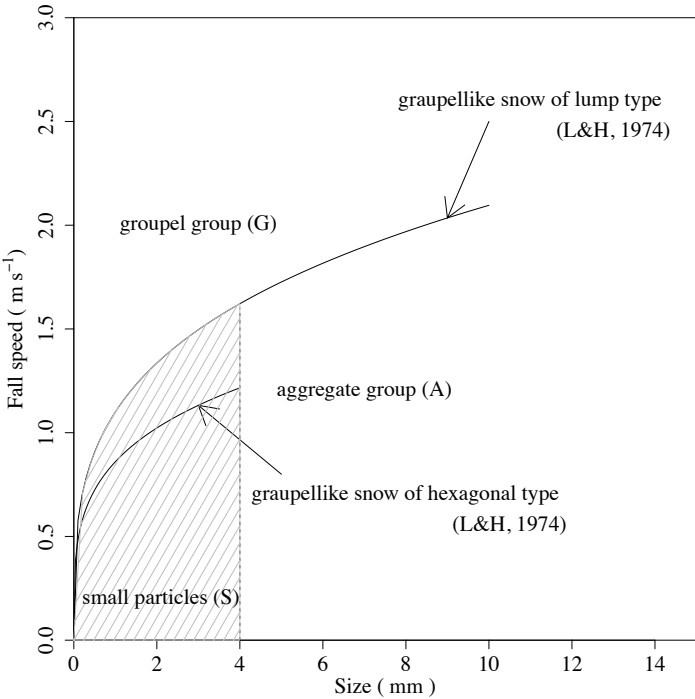

Fig.11. Categories used in snowfall event classification, showing their location in terms of size–fall speed coordinates. The two curves represent size–fall speed relationships for lump-type and hexagonal-type graupels (from Locatelli and Hobbs, 1974, denoted as L&H, 1974 on the graph). The figure was revised from Ishizaka et al. (2016)





Fig. 12. Relationship between SSA of fresh fallen snow and detailed characteristics of falling snow produced by the CMF.
a, b, c: data of aggregate group events in M-type (A group)
d, e, f: data of graupel type events in M-type (G group)
g, h, i: data of small particle type events in M-type (S group)
j, k, i: data in C-type (C group)





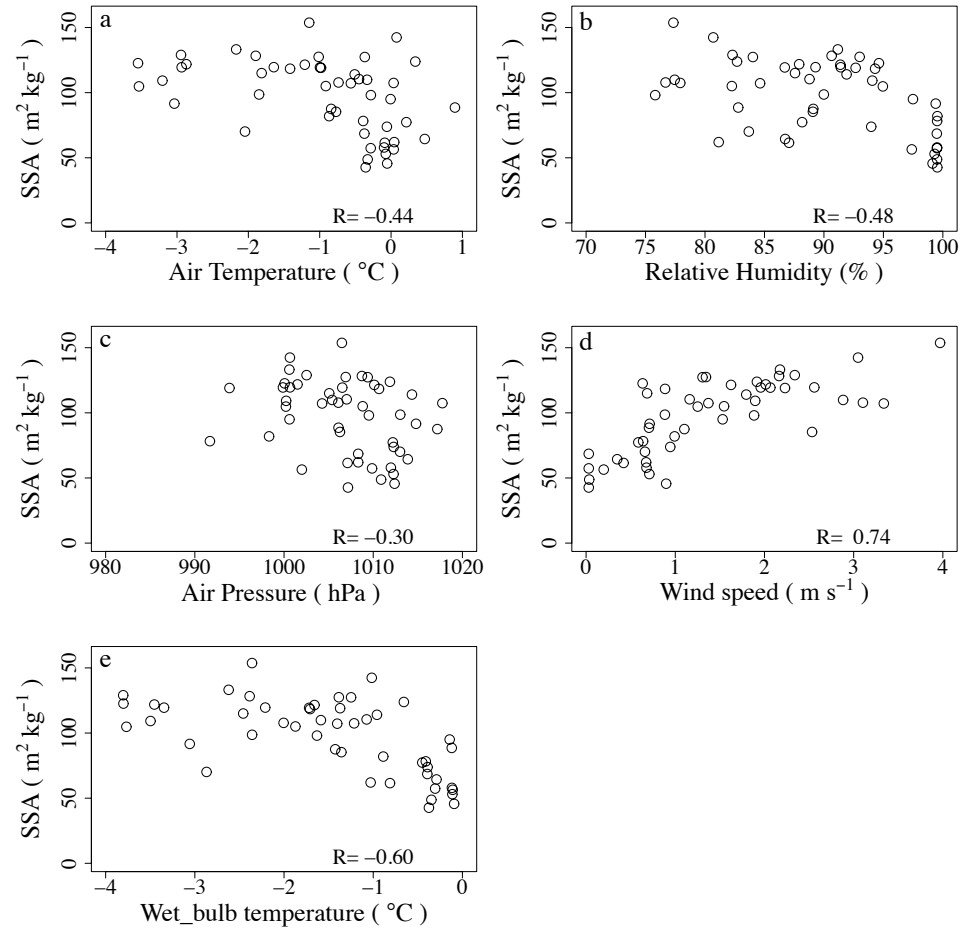

Fig. 13. Relationships between meteorological elements and measured SSA
a: relationship between air temperature and measured SSA
b: relationship between relative humidity and measured SSA
c: relationship between air pressure and measured SSA
d: relationship between wind speed and measured SSA
e: relationship between bulb-wet temperature and measured SSA





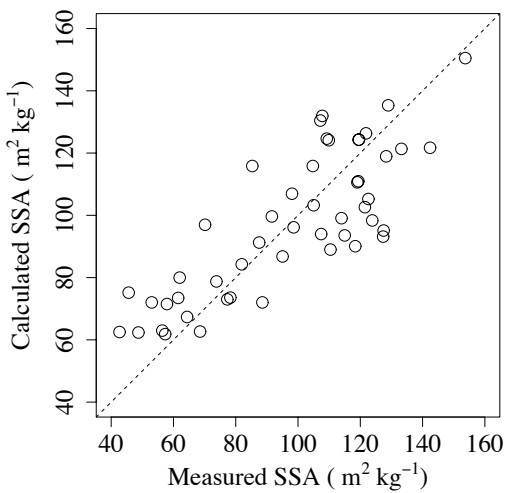

Fig. 14. Comparison between measured SSA and calculated SSA using Eq. (2)
Dotted line indicates 1:1 line





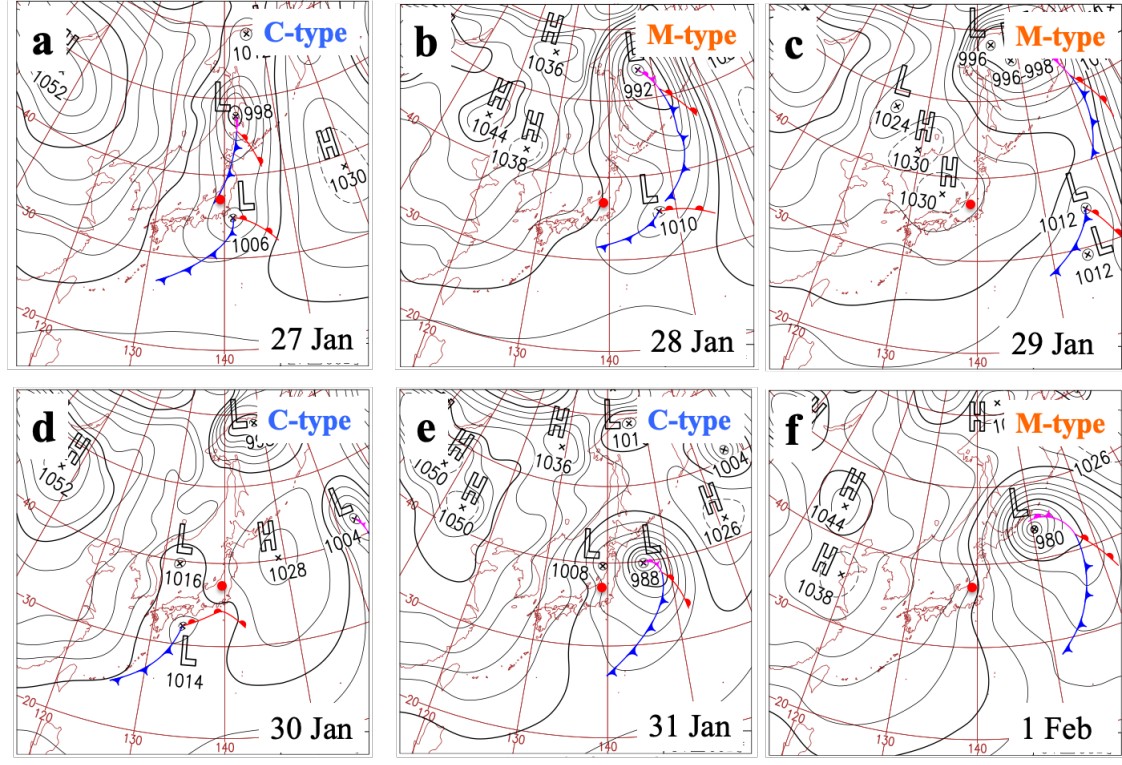

Fig. 15. Synoptic atmospheric pressure condition during the period from Jan 27 to Feb 1, 2015.
Red circles indicate the location of Nagaoka
Each weather chart shows the condition at 09:00. a.m. (JST)
Weather charts were produced by the Japan Meteorological Agency (http://www.data.jma.go.jp/fcd/yoho/hibiten/index.html).




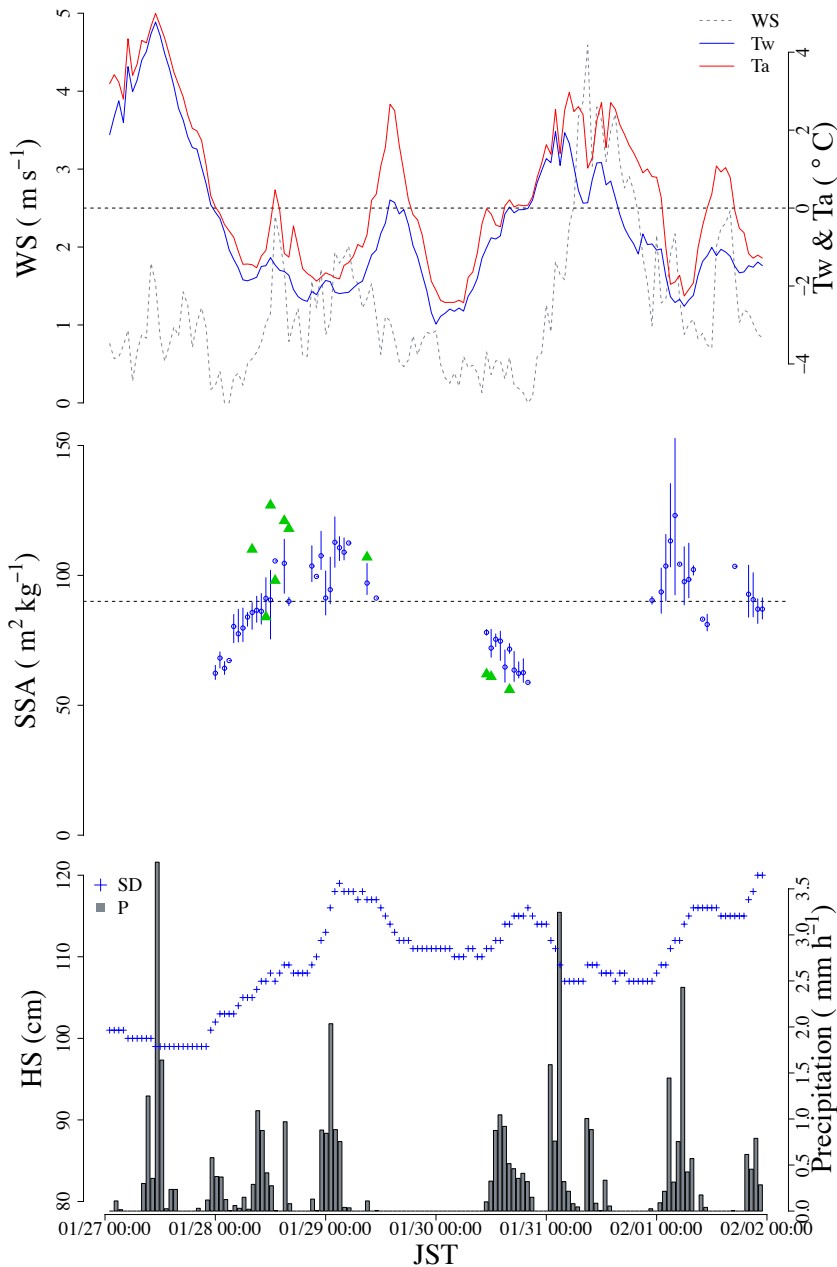

Fig. 16. Simulation results of SSA fluctuations using meteorological data during the period from Jan 27 to Feb 1, 2015 (JST)

WS: wind speed, Tw: wet-bulb temperature, Ta: air temperature, HS: total depth of snow cover, P: Precipitation. Blue bars with circle show the fluctuation of SSA during each 1 h and green triangles show measured SSA





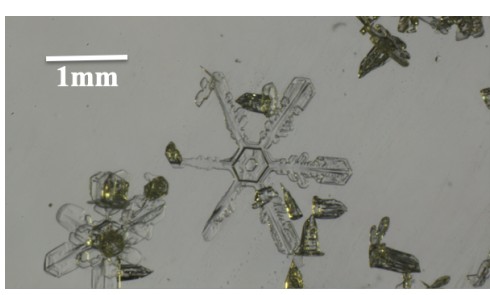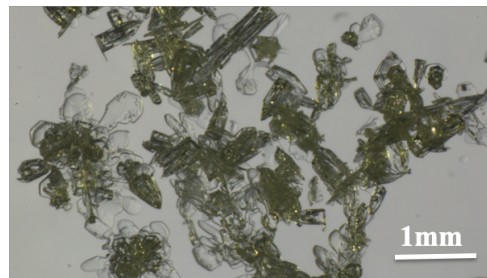

Fig. 17. Unrimed precipitation particles observed at Nagaoka (11:00 a.m. on Jan 30, 2015(JST))