# Peer review of "Measurement of specific surface area of fresh solid precipitation particles in heavy snowfall regions of Japan"

_The Cryosphere, 2019_

## Referee Comment (RC1) · Leena Leppänen (Referee) · 22 May 2019

General comments

The manuscript proposed by Satoru Yamaguchi and others provided interesting description of a study, where properties of precipitation particles were analyzed within 1-2 hours after snowfall events. SSA was observed with methane absorption method and grain type and riming was analyzed from microphotographs. In addition, size and fall speed of falling particles were observed with CCD camera and snowfall modes were defined from Doppler radar measurements. Meteorological parameters were observed to define melting stage and a site specific empirical equation to derive SSA from wind

speed and wet-bulb temperature.

Weak PP layers consisting unrimed crystals is the main reason for weak layers and avalanches in Japan. The study concluded that SSA of fresh PP is influenced by riming, threshold value 90 m2kg-1 was found between unrimed and rimed crystals, and riming degree is related to synoptic weather conditions. The established equation describes fluctuations better than the absolute values of SSA based on meteorological observations, but it is a first step to future development of physical snow model to estimate evolution of weak PP layers for avalanche prediction.

The manuscript is well-structured and presents background, results and conclusions clearly. In addition, comparison with other studies is complete. Following minor comments would improve the manuscript.

Specific comments

Page 2 Line 20: Riming could be described with one sentence

Page 3 Line 13: "over short intervals" could need specification, for example, "over short intervals (1-2 h)"

Page 3 Line 30: "The PP photographs" does it mean CCD camera photos or some other photos like microphotos? Clarify this in the text.

Page 3 Line 32: Figure of the CCD camera system could be nice in Fig. 1.

Page 4 Line 26: Figure of methane absorption device would be nice to be included to Fig.1

Page 4 Line 31: Figure of the SSA sample would be nice in Fig.1

Page 5 Line 4: Microphotography could be described with more details or reference could be added if exists.

Page 5 Line15: References for the equation should be added

Page 5 Line 29: It remains a bit unclear why albedo is mentioned and how the result in Page 6 Line 4 is related to conclusions of this manuscript. My recommendation is to either remove the text or clarify its significance better. In addition, it would be good to compare results with other studies on SSA and NIR albedo/reflectance.

Page 6 Line 21: What means "the selected results"? Please clarify the text.

Page 7 Lines 27-31: Sentences could be moved to Introduction and text could be modified as "The disastrous avalanches in Japan presented in chapter 1 were caused by…"

Page 8 Line 25: "small SSA" could be defined with number

Page 8 Line 29: Chapter 3.4 is more difficult to understand than the other chapters, possibly originating from different types, modes and groups for snowfall which are easily mixed without knowing better the definitions, text could be clarified.

Page 9 Line 9: Why only UFE data with similar trend as no melt events (which have larger SSA than melt events but can still include small SSA values, Page 6 Lines 32-33) is used? It is good to have uniform PP type and no melting? One sentence about this could be added for clarification.

Page 9 Line 26: Could you describe the trend with few words?

Page 11 Line 22: "empirical parametrization" How this parametrization was formed, could be described with one sentence.

Page 13 Line 15: "especially due to the introduction of wind speed in the parameter", which parameter? clarify in the text.

Figure 1d: Is "falling snow crystal photos" CCD camera photos or microphotos?

Figure 2a caption: Add "Optical grain radius is calculated for the data sets by using Eq. (1)."
Figure 12: It would be good to have line for SSA value 90 also in the first column and explanation for the lines needs to be added to the caption.

Figure 13: Could you add lines fitted to the points? It would show more clearly increase or decrease of SSA with meteorological data.

Technical corrections

Page 1 Line 5: Vincent Vionnet, I would assume

Page 1 Line 19: Replace "sometimes" with "have potential to"

Page 1 Line 20: Replace "It is" with "Weak PP layers are"

Page 1 Line 21: "weak PP layer" could be also "those layers"

Page 1 Line 27: "degree of riming of PP" could be "degree of PP riming"

Page 2 Line 7: Consider to remove "of snow cover"

Page 2 Line 12: Replace "sometimes" with "can" or "have potential to" or "occasionally"

Page 2 Line 20: Remove ","

Page 2 Line 21: Replace "because its initial density is small" with "due small initial density"

Page 3 Line 1: Remove " 's"

Page 3 Line 5: Replace "degree of riming of PP" with "degree of PP riming"

Page 3 Line 10: Add "e.g. in the Crocus snowpack model"

Page 3 Line 11: Remove "treats that"

Page 3 Line 11: Replace "maximal value of SSA of PP" with "maximum SSA value of PP"

Page 3 Line 15: Replace "on these data" with "on these parameters"
Page 3 Line 25: Replace "greater" with "larger"

Page 3 Line 30: "windless conditions"

Page 3 Line 30: Consider to remove "in winter"

Page 4 Line 6: Replace "The authors of the current paper..." with "The detailed characteristics of falling snow produced by the CMF was used when..."

Page 4 Line 15: "...for estimating the SSA with an empirical equation."

Page 4 Line 17: Remove "the SIRCS's"

Page 4 Line 30: Remove "new"

Page 4 Line 31: Replace "but" with "however"

Page 4 Line 32: Replace "Because of" with "Due to"

Page 5 Line 1: Sentences "The measures SSAs..." and the next one could be moved before the previous sentence "Because of".

Page 5 Line 8: Add "For the methane absorption method, the averaged heat of adsorption ..."

Page 5 Line 10: Replace "our results" with "results of this study"

Page 5 Line 12: "The figure also shows the optical radius (Ropt) for the data sets..."

Page 5 Line 29: Is "more than three-fold" correct expression?

Page 5 Line 33: Replace "impurities-free" with "impurity free"

Page 6 Line 9: Replace "sometimes" with for example "partly" or "occasionally" or "mostly"

Page 6 Line 23: Remove "one"

Page 6 Line 32: Remove "an"

Page 7 Line 10: Replace "sixty" with "60"

Page 7 Line 11: Replace "Figure 6 is the results of…" with for example "Figure 6 presents the results from…"

Page 7 Lines 14-15: Rephrase as "aggregation and riming to be predominant (Fig. 5c)."

Page 7 Line 31: Replace "All of these avalanches" with "The avalanches"

Page 7 Line 33: Replace "introduction of information of PP type to numerical" with "description of PP type information in numerical"

Page 8 Line 3: Replace "To investigate the measurements" with "When the measurements under M-type are investigated…"

Page 8 Line 7: Replace "fifty-three" with "53"

Page 8 Line 8: Replace "sets of data" with "data sets"

Page 8 Line 11: Replace "For more information" with for example "for information of grain shape and riming"

Page 8 Line 21: It is bit confusing to have snowfall types and modes defined both with letters (including M-type and M mode). I would recommend to use full names for monsoon-type and cyclone-type.

Page 8 Line 23: Replace "classification of the degree of riming" with "classification of the riming degree"

Page 8 Line 33: Replace "Ishizaka et al. (2016)'s CMF distribution" with "CMF distribution of Ishizaka et al. (2016)"

Page 9 Line 3: "In such case, these…"

Page 9 Line 7: "49"

Page 9 Line 8: "Figure 10"

Page 9 Line 9: Replace "became" with "is"

Page 9 Line 9: Replace "its" with "UFE"

Page 9 Line 11: "Figure 6"

Page 9 Line 11: "…synoptic scale condition, and… "

Page 9 Line 11 and 12: Remove "its"

Page 9 Line 14: It is confusing to have S mode and S particle group. I would recommend to use for example "AGG", "GRA" and "SMA" groups.

Page 9 Line 15: Remove "in their paper"

Page 9 Line 16: Replace "the authors in this study treated" with "this study treated"

Page 9 Line 20: Replace "Third" with "The third physical characteristic"

Page 9 Line 21: Replace "five-min" with "five minute"

Page 9 Line 23: Consider to remove "(p)"

Page 10 Line 2: Replace "the reason D and SSA in the G group are related" with for example "the reason for relationship between snowfall mode D and SSA in the M-type snowfall group G "

Page 10 Line 7: "classification of riming degree"

Page 10 Line 12: "graupelike" to "graupellike"

Page 10 Line 18: Remove "their"

Page 11 Line 24: "the valid range in Eq. (2)"

Page 11 Line 33: Replace "periodically changed" with "alternated"

Page 12 Line 22: Remove "calculation"

Page 12 Line 23: Replace "In fact, the authors of this study" with "In fact, the unrimed PPs were observed in Nagaoka on Jan 30, 2015 as expected (Figure 17)."

Page 12 Line 32: "from" to "between"

Page 13 Line 12: "...derived. This equation..."

Page 13 Line 12: "...M-type..."

Page 13 Line 14: Replace "Thus, although it has its limits stemming from" with "Although, regardless of limiting site-dependent parametrization"

Page 13 Line 19: Remove "its"

Figure 5 caption: Two first sentences needs "." in the end.

Figure 6 caption: "fresh PP"

Figure 7 caption: Replace "Each mode is shown in Table 1." with "Each mode is described in Table 1."

Figure 9 caption: Remove "." from "1 min." in the first row.

Figure 12 caption: "C-type (C type)"

Figure 16: Legend has "SD" instead of "HS"

Figure 16 caption: To the last row could be added "...the fluctuations of simulated SSA during..."

---

## Referee Comment (RC2) · Anonymous Referee #2 · 7 Jun 2019

**Review of the manuscript**

**Manuscript title**

Measurement of specific surface area of fresh solid precipitation particles in heavy snowfall regions of Japan

**General comments**

Within the manuscript, measurements of the specific surface area (SSA) of precipitation particles (PP) using the methan adsorption method are presented for a heavy snowfall region in Japan during four winters between 2013 and 2017. The influence of melting events, the synoptic meteorological condition, the degree of riming, as well as meteorological parameters measured at the observation site on the SSA are investigated. After careful classification of the snow samples, a parameterization of the SSA with respect to wind speed and wet-bulb temperature is derived. In reproducing observed fluctuations of the SSA with this parameterization, the study aims to provide a first step in describing the development of weak PP layers within snow cover models which is essential for avalanche forecasting. Even though the parameterization is site-dependent, the methodology can be readily applied to other observation sites.

The manuscript is clearly structured and the great amount of work going into analyzing and selecting suitable snow samples for the development of the parameterization should be acknowledged. Overall, the figures are of good quality and help to convey the arguments of the authors. However, there are some aspects that need further focus in my opinion. After some general comments, the more specific comments follow below.

A detailed discussion of the measurement uncertainties connected with the SSA measurements is missing. The statistical variations along the different snow samples need to be clearly separated from instrumental uncertainties.

Section 3 follows many different steps of selection of snow samples for the final parameterization. A separate subsection within Section 2 (Methodology) where the individual steps (melting, synoptic situation, CMF analysis...) are introduced is highly recommended.

Most of the time, the figures were just referenced within the text without being described in more detail first (e.g., Fig. 9a at page 9). At the beginning, each figure should be explained and described before being discussed. This will definitely foster reading comprehension if changed throughout the manuscript.

At some points, difficult sentence constructions and the use of English prevent fluent reading. Within the technical corrections at the end of this review, some typing errors and suggestions for re-formulating are stated. Please consider spending some more time on proof reading the manuscript as this would improve its clarity.

**Specific comments**

**Introduction**

P2 Line 7: For these reasons... please revise the sentence, the meaning is not clear.

**Results and discussion**

P5 Lines 16-26: The whole paragraph needs revision. More detail is needed concerning the chosen data samples by Domine et al. and Schleef et al (sampling strategy, observation site,...). The paragraph raises questions whether these datasets are comparable at all (other region, variability in PP, ...).

Figure 2: specify the figure caption, Measurement of SSA and Ropt, explain the different symbols within the plots

P6 Line 2: This is due to the spectral behaviour of the imaginary part of the complex refractive index of ice and should be stated here.

P6 Line 3: are these integrated values for the albedo? Please give the spectral range corresponding to the albedo values.

P6 Line 33: Is this really just the melting effect? SSA samples of different days are used, so different meteorological conditions within the clouds and the atmosphere will influence the SSA of fresh PP anyway, even without melting. You demonstrate this yourself later in this section.

P6 Line 27: move this paragraph to the introduction?

P9 Line 3: How are size and fall speed measurements done? Within Section 2 (Methodology), you just mention the CCD camera system and reference Ishizaka et al. (2004). It has to be included that the size corresponds to the maximum horizontal width.

Figure 10: The values illustrated by the gray box plot and the mean value of NME in Figure 10 are not the same as in Figure 3. Why is this the case?

Figure 11: adding sampled data and demonstrating the characterization within this plot would be more convincing.

P9 Line 25: Ropt decreases with increasing SSA. Add within the discussion that D and Ropt are different.

P9 Line 26: why is C-type not classified due to PP type?

P10 Lines 10-19: this discussion is doubtful as the different trends stated within the manuscript for values below and above 90 m2 kg-1 are solely dependend on one measurement (lowest SSA).

P11 Line 24: Add the percentage of snowfall events at which Eq. 2 is applicable with respect to wind speed and Tw.

P12 Line 2: For the development of Eq. (2), 1-min meteorological data was used (as explained in Section 3.2). Why do you switch to 10-min meteorological data now?

**Technical corrections**

P1 Line 24: gas adsorption method

P1 Line 24: font size of PP is larger than in the rest of the text

P2 Line 14: LaChapelle

P4 Line 8, 19, and 21: delete whitespace at section reference

P4 Line 11: radiation, no plural

P4 Line 12: snow water equivalent, no plural

P4 Line 17: gathers information on snow clouds during

P4 Line 27: use {[(...)]} bracket convention, and include the reference in the bracket

P4 Line 29: measurement interval

P4 Line 30: unit is ml, not mL

P5 Line 11: no brackets around sentence: Hereafter, data ...

P5 Line 27: winter instead of fourth season

P6 Line 1: NIR, explain acronym here, not in Line 3

P6 Line 1: show the albedo at the near-infrared (NIR) wavelengths is affected more significantly by the change in SSA than the albedo at the visible wavelength range.

P6 Line 4: These results indicate that the information on SSA variation of fresh PP is important for the simulated evolution of the local surface radiation budget.

P6 Line 11: adjust the hyphen

P6 Line 25: PP falling at T

P6 Line 32: that fresh PP can have a small SSA

P7 Line 32: To predict... : please split this sentence for better reading comprehension.

P8 Line 8: L mode

P8 Line 9: could not be determined

P9 Line 8: wrong reference, Figure 9 should be Figure 10

Figure 11: label for group (G) should be graupel group, not groupel group

P9 Line 21: use the symbol $\rho$ for density

P9 Line 22: the initial deposited... the meaning of this part of the sentence is unclear.

P9 Line 32: This is only a hypothesis, ...

P11 Line 6: use subscript for Ta

P11 Line 7: surface air pressure, also: use physical symbol $p$

P12 Line 7: in the liquid phase

P12 Line 21: values show some difference

P13 Line 1: whitespace around hyphen

P13 Line 12: missing '.' after derived.

P13 Line 15: first step towards

P13 Line 20: fresh PP

P13 Line 22: fresh PP

---

## Editor Comment (EC1) · Florent Dominé (Editor) · 3 Jul 2019

Dear Authors

The discussion is now closed. Please note the new procedure: before submitting a revision you must first address the reviewers' comments. Note that at this stage you do not need to address all comments point by point. Please just respond to the main comments and how you plan to address them in a revised version. You will be allowed to upload a revised version after that stage.

Best regards

[Figure]

Florent Domine Editor

---

## Author Comment (AC1) · 29 Jul 2019

We wish to express our strong appreciation to the reviewer for their insightful comments on our paper. We feel the comments have helped us significantly improve the paper. We submit the supplement, in which we have addressed all the comments and substantially revised the documents.

Please also note the supplement to this comment:
https://www.the-cryosphere-discuss.net/tc-2019-78/tc-2019-78-AC1-supplement.pdf

**Supplement:**

**Responses to the Reviewers' comments**

We would like to express our appreciation to the reviewers for their constructive reviews and helpful advice. We believe that most of reviewers' comments were fair and proper, and we found them very helpful in revising this paper. We have addressed all the comments and substantially revised the documents and trust that all revisions will be satisfactory.

29 July, 2019

Snow and Ice Research Center

National Research Institute for Earth Science and Disaster Resilience

Satoru Yamaguchi

**Responses to Referee 1 (Leena Leppänen)'s comments**

The reviewer's words are given in italics and bold; Parts of the revised texts are shown in <\*\*\*\*\*\*\*\*>.

<Specific comments>

***Page 2 Line 20: Riming could be described with one sentence***

***Page 3 Line 13: "over short intervals" could need specification, for example, "over short intervals (1-2 h)"***

We improved them in the revised paper

***Page 3 Line 30: "The PP photographs" does it mean CCD camera photos or some photos like microphotos? Clarify this in the text.***

PP photographs were taken using a close-up camera and this information is added in the revised paper.

***Page 3 Line 32: Figure of the CCD camera system could be nice in Fig. 1.***

***Page 4 Line 26: Figure of methane absorption device would be nice to be included to Fig 1***

***Page 4 Line 31: Figure of the SSA sample would be nice in Fig.1***

We added Fig.1e (Falling snow particle observation system using CCD camera), 1f (Portable developed device for the methane adsorption method) and 1g (sample folder for measurement SSA) in the revised paper.

***Page 5 Line 4: Microphotography could be described with more details or reference could be added if exists.***

We took microphotography using a microscope, so we added this information in the revised paper.

***Page 5 Line15: References for the equation should be added***

We added the reference (Grenfell and Warren, 1999) in the revised paper.

*Page 5 Line 29: It remains a bit unclear why albedo is mentioned and how the result in Page 6 Line 4 is related to conclusions of this manuscript. My recommendation is to either remove the text or clarify its significance better. In addition, it would be good to compare results with other studies on SSA and NIR albedo/reflectance.*

Albedo is an important parameter to calculate snow surface radiation budget. Albedo depends on physical properties of snow including SSA. Therefore, it is important to understand the influence of fluctuation of SSA of PP on Albedo values. Our results indicate that the albedo at the NIR wavelengths changed with fluctuation of SSA of PP, and that information of SSA variation of fresh PP is important to simulate correct NIR albedo. Therefore, our attempt to establish the function of SSA using meteorological data should contribute to the improvement of albedo calculation scheme in snow cover models. From these points of view, we improved the text as follows:

< These results indicate that the information on SSA variation of fresh PP is important for the simulated evolution of the local surface radiation budget. Therefore, parameterization of SSA fluctuations is essential for accurate simulation of NIR albedo in natural snow. >

*Page 6 Line 21: What means "the selected results"? Please clarify the text.*

The selected results indicate "melt events" and "no melt events" classified using $T_w$. We improve the text as follows:

< Figure 3 shows the classified results (ME and NME) using $T_w$.>

*Page 7 Lines 27-31: Sentences could be moved to Introduction and text could be modified as "The disastrous avalanches in Japan presented in chapter 1 were caused by..."*

We moved these sentences to the introduction.

*Page 8 Line 25: "small SSA" could be defined with number*

We added the specific values ($< 80 \text{ m}^2 \text{ kg}^{-1}$) in the text

***Page 8 Line 29: Chapter 3.4 is more difficult to understand than the other chapters, possibly originating from different types, modes and groups for snowfall which are easily mixed without knowing better the definitions, text could be clarified.***

We improved the texts as follows:

< As shown in Figure 6, the SSA strongly depends on its synoptic scale condition; therefore, the relationship between SSA of fresh PP and detailed characteristics of PP should also depend on synoptic scale conditions. For this reason, firstly the UFE data were classified into the two synoptic scale conditions (M-type and C-type). In addition, M-type data were classified into three groups based on the PP types [aggregate group (AGG), graupel group (GRA), and small particle group (SMG)] using CMF analyses reported by Ishizaka et al. (2016) (Fig. 11). Although Ishizaka divided a small particle group (SMG) into two subgroups (S1 and S2), this study treated S1 and S2 as one group (SMG) (Fig. 11). Finally, four data groups (C-type, AGG, GRA, and SMG) were used for analyses. >

Moreover, we show the list of data set for each section as Table 1 as follows:

Table 1 List of data set

| Name | Condition | Sample number | Section |
|---|---|---|---|
| All data | Data including all measured data | 102 | Section 3.1 |
| *no melt events* (NME) | Data without melting in All data. Web-bulb temperature (Tw) <0 °C | 72 | Section 3.2 Section 3.3 |
| *melt events* (ME) | Data with melting in All data. Web-bulb temperature (Tw)$\geqq$ 0 °C | 30 | Section 3.2 |
| *uniform fallings event* (UFE) | Data with a single PP type during the deposition period in NME | 49 | Section 3.4 Section 3.5 |

***Page 9 Line 9: Why only UFE data with similar trend as no melt events (which have larger SSA than melt events but can still include small SSA values, Page 6 Lines 32-33) is used? It is good to have uniform PP type and no melting? One sentence about this could be added for clarification.***

We improved the text as follows:

< CMF distributions of all cases in NME were graphed and inspected visually based on these analyses, forty-nine UFE were selected from NME (Table 1).>

We also added the Table 1 showing the list of used data sets for each section

***Page 9 Line 26: Could you describe the trend with few words?***

We added the following sentence:

< Basically, the trends of GRA and SMA remain the same, while those of AGG are different from the other two groups. >

***Page 11 Line 22: "empirical parametrization" How this parametrization was formed, could be described with one sentence.***

We added the following explanation:

<Equation (2) is this parameterization with the least squares method:>

***Page 13 Line 15: "especially due to the introduction of wind speed in the parameter", which parameter? clarify in the text.***

We added the following explanation:

  <the parameter of the empirical equation>

***Figure 1d: Is "falling snow crystal photos" CCD camera photos or microphotos?***

We added the following explanation:

< snow crystal photos on the belt conveyer with a close-up camera>

***Figure 2a caption: Add "Optical grain radius is calculated for the data sets by using Eq. (1)."***

We added this sentence in the revised paper.

***Figure 12: It would be good to have line for SSA value 90 also in the first column and explanation for the lines needs to be added to the caption.***

We added the lines and the explanation in the revised paper.

***Figure 13: Could you add lines fitted to the points? It would show more clearly increase or decrease of SSA with meteorological data.***

We added the linear approximation line in the revised paper

<Technical corrections>
All of them were done, and then the improved texts were corrected by a native English speaker.

The reviewer's words are given in italics and bold; Parts of the revised texts are shown in <********>.

<General comments>

***A detailed discussion of the measurement uncertainties connected with the SSA measurements is missing. The statistical variations along the different snow samples need to be clearly separated from instrumental uncertainties.***

As shown in the text, uncertainties of SSA measurement [measurement repeatability (standard deviation)] as 3%, therefore, the SSA differences shown in section 3.2 (between "melt events" and "no melt events") and section 3.3 (between "C-type" and "M-type" and SSA difference between different modes) are statistically significant. We also added the accuracy information of each meteorological equipment to section 2.1 of the revised paper. Considering the accuracy of each meteorological equipment, the discussions in section 3.5 and 3.6 are also statistically significant. On the other hand, the relationship between SSA and averaged fall speed ($V$): initial density ($\rho$) of Aggregate group change trends at the border of 90 $m^2$ $kg^{-1}$, which are discussed in section 3.4, may be doubtful from the statistical view point because of small sample numbers with the consideration of instrumental uncertainties. For this reason, we eliminated the discussion part of the relationship between SSA and averaged fall speed ($V$), initial density ($\rho$) of Aggregate group. Then we added the following sentence in section 3.4 of the revised paper.

< The relationship between SSA and D and $\rho$ of AGG seem to be more complex than other two groups. These results may be resulting from the different degrees of riming on the crystal.>

***Section 3 follows many different steps of selection of snow samples for the final parameterization. A separate subsection within Section 2 (Methodology) where the individual steps (melting, synoptic situation, CMF analysis...) are introduced is highly recommended.***

We added section 2.3 "Data selection" and Table 1. In section 2.3 and Table 1, we summarized which data sets were used in each section as follows:

<2.3 Data selection
In the study, several selected data sets (Table 1) were provided for aim of each analysis. In the discussion of general characteristics of SSA of PP in Nagaoka (Section 3.1), all measured data

were used. In the discussion of influence of melting effect (Section 3.2), all measured data were classified into two data sets: no melt events (NME) affected by no melt effect and melt events (ME) affected by melt effect, and then discussed. In the discussion of the relationship between SSA and synoptic meteorological conditions (Section 3.3), only NME was used. In Section 3.4 and 3.5, uniform fallings event (UFE), in which only data taken under a single PP type condition during the deposition period was selected from NME, and used for discussion. Detail information of data selection conditions are shown in each section. >

Table 1 List of data set

| Name | Condition | Sample number | Section |
|---|---|---|---|
| All data | Data including all measured data | 102 | Section 3.1 |
| *no melt events* (NME) | Data without melting in All data. Web-bulb temperature (Tw) $<0\,°C$ | 72 | Section 3.2 Section 3.3 |
| *melt events* (ME) | Data with melting in All data. Web-bulb temperature (Tw)$\geqq$ $0\,°C$ | 30 | Section 3.2 |
| *uniform fallings event* (UFE) | Data with a single PP type during the deposition period in NME | 49 | Section 3.4 Section 3.5 |

***Most of the time, the figures were just referenced within the text without being described in more detail first (e.g., Fig. 9a at page 9). At the beginning, each figure should be explained and described before being discussed. This will definitely foster reading comprehension if changed throughout the manuscript.***

We added the following sentence in the text for the explanation of Fig. 9.

<Figure 9 shows the representative CMF plots under different conditions.>

We check the whole texts and improved them as needed.

***At some points, difficult sentence constructions and the use of English prevent fluent reading. Within the technical corrections at the end of this review, some typing errors and suggestions for re-formulating are stated. Please consider spending some more time on proof reading the manuscript as this would improve its clarity.***

Before re-submitted, the revised texts were corrected by a native English speaker.

<Specific comments>

*P2 Line 7: For these reasons... please revise the sentence, the meaning is not clear.*

We revised the text as follows:

< To simulate accurate continuous change of the physical properties of snow, the formulation of temporal variations in the SSA are important.>

*Figure 2: specify the figure caption, Measurement of SSA and Ropt, explain the different symbols within the plots*

We improved the Figure 2 as following, in which different colors were used for SSA and R$_{opt}$:

[Figure]

*P6 Line 2: This is due to the spectral behaviour of the imaginary part of the complex refractive index of ice and should be stated here.*

We added the above sentence in the revised paper

*P6 Line 3: are these integrated values for the albedo?*

*Please give the spectral range corresponding to the albedo values.*

The integrated values for the shortwave albedo are from 0.87 to 0.90. The boundary of the visible and NIR spectral domains is 0.7 μm. This information was added in the revised paper as follows:

< In fact, the UV-visible (wavelength = 0.2-0.7 μm) albedo value simulated using the measured maximum and minimum optical radii show almost the same values (0.99), while the simulated NIR (wavelength = 0.7-3.0 μm) albedo values vary from 0.75 to 0.80.>

*P6 Line 33: Is this really just the melting effect? SSA samples of different days are used, so different meteorological conditions within the clouds and the atmosphere will influence the SSA of fresh PP anyway, even without melting. You demonstrate this yourself later in this section.*

We agree the reviewer's comment, namely, different meteorological conditions within the clouds and the atmosphere will influence the SSA of fresh PP, therefore, we improved the text as follows:

< These results indicate that fresh PP can have a small SSA even without melting, and other factors, such as meteorological conditions within the clouds and the atmosphere, for controlling SSA of fresh PP ought to be considered. >

*P7 Line 27: move this paragraph to the introduction?*

We moved this paragraph to the introduction.

*P9 Line 3: How are size and fall speed measurements done? Within Section 2 (Methodology), you just mention the CCD camera system and reference Ishizaka et al. (2004). It has to be included that the size corresponds to the maximum horizontal width.*

We added the above information in section 2.1 as follows:

< Additionally, the characteristics of falling snowfall particles, including size and fall speed, are automatically measured using a CCD camera system (Fig. 1e) with particle size resolution of 0.25 mm for width and 0.50 mm for height (Ishizaka et al., 2004)>

***Figure 10: The values illustrated by the gray box plot and the mean value of NME in Figure 10 are not the same as in Figure 3. Why is this the case?***

We made the mistake to put a label on the figure. We corrected the label in the revised paper.

***Figure 11: adding sampled data and demonstrating the characterization within this plot would be more convincing.***
***P9 Line 26: why is C-type not classified due to PP type?***

CMF method was basically developed using data under M-type condition. Therefore, it is difficult to classify PP type under C-type using CMF (Please see the following figure). Therefore, we only used CMF for classification of PP type under M-type condition. For this reason, we only plotted the data under M-type condition in Fig 11.

[Figure]

Relationship between CMF and four groups (GRA, AGG, SMC, and C-type)

***P9 Line 25: Ro decreases with increasing SSA. Add within the discussion that D and Ro are different.***

We added the above sentence in the revised paper.

***P10 Lines 10-19: this discussion is doubtful as the different trends stated within the manuscript for values below and above 90 m² kg⁻¹ are solely dependend on one measurement (lowest SSA).***

As already mentioned in the response to general comments, we agree that the different trends for values below and above 90 $m^2$ $kg^{-1}$ of Aggregate group is doubtful. Therefore, we eliminated this discussion in the revised paper.

***P11 Line 24: Add the percentage of snowfall events at which Eq. 2 is applicable with respect to wind speed and Tᵥ.***

We analyzed data in 2012 winter season to investigate the adaptable ratio to the valid range in Eq. (2) during winter season in Nagaoka using 1-h resolution meteorological data. The analyses results show that 84% snowfall events were in the valid range of Eq. (2) while 13 % snowfall events were not in the valid range because $Tw \geqq 0$ °C. Based on these results, we added the following sentence:

< To investigate the adaptable ratio to valid range in Eq. (2) during winter in Nagaoka, the data of 2015 winter (December 2014 - March, 2015) were analyzed using meteorological data with 1 h resolution: 445 snowfall events occurred during the winter of 2015 (here, snowfall event is defined as where the snow height increased during 1 h) and 374 cases of 445 snowfall events (84% cases of snowfall events) were in the valid range of Eq. (2). In the case of outrange in Eq. (2), Tw $\geqq$ 0 ° C (59 cases) and WS > 4 m s-1 (14 cases) (the case of Tw $\geqq$ 0 ° C & WS > 4 m s-1 :3 cases). Therefore, there is still room for improvement to treat SSA under melting effect simulation in Eq. (2).   >

***P12 Line 2: For the development of Eq. (2), 1-min meteorological data was used (as explained in Section 3.2). Why do you switch to 10-min meteorological data now?***

Although the basic data is 1-min resolution, the data for development of Eq. (2) were averaged data over the period with falling snow during the sample period. Therefore, the actual time resolution of data for Eq. (2) should be much longer than 1-min. Influence of PP type variation should be neglected if the time resolution of SSA calculation was sufficiently short. As mentioned in the paper, 1-h resolution is sometimes too long to avoid the influence of PP type variation. On the other hand, 1-min resolution is too short if we introduce our SSA simulation scheme to the snow cover model. We consider that 10-min time resolution is enough short to

have same PP type, and 10-min time resolution will not become a problem when our SSA simulation scheme is installed into the snow cover model. For these reasons, finally we adopted the 10-min resolution meteorological data to calculate SSA in this study.

<Technical corrections>
*Results and discussion*
***P5 Lines 16-26: The whole paragraph needs revision. More detail is needed concerning the chosen data samples by Domine et al. and Schleef et al (sampling strategy, observation site,...). The paragraph raises questions whether these datasets are comparable at all (other region, variability in PP, ...).***

We added Table 2 including the measurement method and observation site information about the data of Domine et al.2007, and Schleef et al. 2014 as follows. Moreover, we eliminated 16 data from the data set of Schleeef et al. 2014 in the analyses of the revised paper, because they were measurement data of artificial snow. Therefore, we used 8 data from the data set of Schleef et al. 2014 and 68 data of Domine et al. 2007, for analyses.

Table 2 Summary of data sets in Domine et al. (2007), Schleef (2014)

| Name | Measurement method | Observation sites |
|---|---|---|
| Dom2007 (Domine et al, 2007) | methane gas adsorption method | French Alps, Arctic, Alaska |
| Sch2014 (Schleef, 2014) | X-ray microtomography | Davos in Swiss |

Other comments were done, and then the improved texts were corrected by a native English speaker.

---

## Author Comment (AC3) · 29 Jul 2019

We wish to express our strong appreciation to the reviewers for their insightful comments on our paper. We feel the comments have helped us significantly improve the paper. We submit the supplement in which we have addressed all the comments and substantially revised the documents.

Please also note the supplement to this comment:
https://www.the-cryosphere-discuss.net/tc-2019-78/tc-2019-78-AC3-supplement.pdf

---

## Author Comment (AC4) · 29 Jul 2019

[revised manuscript text omitted]

ACKNOWLEDGEMENT

We would like to acknowledge the members of SIRC for their participation in useful discussions. Helpful comments and suggestions from L. Leppänen and one anonymous reviewer are greatly appreciated. 
[revised manuscript text omitted]

---

## Author Response (AR1)

**Responses to the Editor's comments**

We would like to express our appreciation to you for your helpful advice. We have addressed all the comments and substantially revised the documents and trust that all revisions will be satisfactory. Finally, the improved texts were corrected again by a native English speaker.

28 August, 2019

Snow and Ice Research Center

National Research Institute for Earth Science and Disaster Resilience

Satoru Yamaguchi

**Responses to comments**

The editor's words are given in italics and bold; Parts of the revised texts are shown in <********>.

***Page 4, paragraph 2. Are the numbers in parentheses measurements uncertainties? This is not clear. Units for radiation do not make sense. You should not use instrument sensitivity but actual uncertainty in W m-2.***

We corrected this paragraph to show the uncertainty for each measurements as follows

<The SIRC also acquires standard meteorological measurements [air temperature ($\pm 0.1℃$) and relative humidity ($\pm 2\%$) at 3.5 m above ground level, wind speed ($\pm 0.3$ m s-1) and wind direction ($\pm 3°$) at 8.7 m above ground level, precipitation (3%) at 3.1 m above ground level, incoming and outgoing shortwave ($\pm 10$ W m-2)/longwave ($\pm 10$ W m-2) radiations at 6.5 m above ground level, air pressure ($\pm 0.35$hPa) measured at 3.0 m above ground level, and surface temperature ($\pm 0.5℃$), snow height ($\pm 1.5$ cm), and snow water equivalents ($\pm 10$ mm)] at various time resolutions (1 min, 10 min, and 1 h) on field>

***Page 5, line 11. "for aim of each analysis". Not clear, please reword.***

We eliminated the term "aim of" from the revised text for escape ambiguous expression as follows:

< In the study, several selected datasets (Table 1) were provided for each analysis.>

***Page 14, line 10. Replace "Colonne" with "Calonne".***

We correct her name not only in the reference but also in the text.

***Table 2. Under Dom2007, replace "French Alps, Arctic, Alaska" with "French Alps, Arctic Canada, Alaska, Svalbard". Under Sch2014, replace "Swiss2 with "Switzerland".***

We revised table 2 based on your advice.

*Figure 1 caption. Replace "belt conveyer" with "conveyor belt". Same in main text. Replace "Potable developed device" with "Portable device developed".*

We correct this term in the capture and text.

*Please check Figure 13 caption. Should "Broken lines are results of linear approximation" be replaced with "Dashed lines are linear least-square fits"?*

We corrected the expression based on your advice.

[revised manuscript text omitted]

---

## Editor Decision (ED1)

Dear Authors,

Thank you for the thoroughly revised version of your manuscript on fresh snow SSA. I am ready to accept your paper pending the minor modifications listed below. I also noticed a number of errors concerning the English language, so please have once again your text checked by a native English speaker.

Thank you for your contribution to The Cryosphere.

Best regards,

Florent Domine

Co-Editor-in-Chief

Page 4, paragraph 2. Are the numbers in parentheses measurements uncertainties? This is not clear. Units for radiation do not make sense. You should not use instrument sensitivity but actual uncertainty in W m$^{-2}$.

Page 5, line 11. "for aim of each analysis". Not clear, please reword.

Page 14, line 10. Replace "Colonne" with "Calonne".

Table 2. Under Dom2007, replace "French Alps, Arctic, Alaska" with "French Alps, Arctic Canada, Alaska, Svalbard". Under Sch2014, replace "Swiss2 with "Switzerland".

Figure 1 caption. Replace "belt conveyer" with "conveyor belt". Same in main text. Replace "Potable developed device" with "Portable device developed".

Please check Figure 13 caption. Should "Broken lines are results of linear approximation" be replaced with "Dashed lines are linear least-square fits"?